# Tailoring the component of protein corona via simple chemistry

Xiang Lu[1,2,5], Peipei Xu[3,5], Hong-Ming Ding[2]*, You-Sheng Yu[1], Da Huo[4]* & Yu-Qiang Ma [1]*

Control over the protein corona of nanomaterials allows them to function better. Here, by taking graphene/gold as examples, we comprehensively assessed the association of surface properties with the protein corona. As revealed by in vitro measurements and computations, the interaction between graphene/gold and HSA/IgE was inversely correlated with the hydroxyl group availability, whereas the interaction between that and ApoE was comparatively less relevant. Molecular simulations revealed that the number and the distribution of surface hydroxyl groups could regulate the manner in which nanomaterials interact with proteins. Moreover, we validated that ApoE pre-adsorption before injection enhances the blood circulation of nanomaterials relative to their pristine and IgE-coated counterparts. This benefit can be attributed to the invulnerability of the complementary system provided by ApoE, whose encasement does not increase cytotoxicity. Overall, this study offers a robust yet simple way to create protein corona enriched in dysopsonins to realize better delivery efficacy.

[1] National Laboratory of Solid State Microstructures and Department of Physics, Collaborative Innovation Center of Advanced Microstructures, Nanjing University, Nanjing, Jiangsu, China. [2] Center for Soft Condensed Matter Physics and Interdisciplinary Research, School of Physical Science and Technology, Soochow University, Suzhou, Jiangsu, China. [3] Department of Hematology, Drum Tower Hospital, School of Medicine, Nanjing University, Nanjing, Jiangsu, China. [4] School of Pharmacy, Nanjing Medical University, Nanjing, Jiangsu, China. [5]These authors contributed equally: Xiang Lu, Peipei Xu. *email: dinghm@suda.edu.cn; huoda@njmu.edu.cn; myqiang@nju.edu.cn

Nanomaterials have been extensively explored for biomedical applications, such as drug delivery and early-stage disease diagnosis[1-5]. The relatively slow translation of nanomedicine into clinical applications has encouraged researchers to pursue the yet unknown factor behind the failure of many nanomedicines[6-9]. One major limitation of engineered nanomaterials is their undetermined fate in vivo, which is revealed to be associated tightly with the protein corona effect[6,8]. Principally, shortly after entry into the blood, the outer surface of nanomaterials become passivated by numerous biological species, the dominant type of which is serum protein. This is where the term protein corona effect got the name[10]. The protein corona substantially remodels the nanomaterials' manner of interaction with cells[11-14]. Therefore, an in-depth understanding of the protein corona effect can effectively facilitate the translation of nanomedicines. To date, most efforts have been devoted to understanding the composition of protein coronas as well as their biological implications[15]. For instance, Dawson and co-workers[16] nicely uncovered mechanism underlying the protein corona-targeting ligand interaction. Tenzer et al.[17] showed that the protein corona can be established within minutes of exposure to body fluid-mimicking medium, which led to drastic pharmacokinetic changes that also determine the cytotoxicity of the nanoparticles.

In an effort to mitigate the complications caused by the protein corona effect, initial attempts wherein anti-fouling coating was utilized were made. The best known polymer developed to this end is poly(ethylene glycol) (PEG)[18-20]. In addition, zwitterionic polymers such as polybetaines and polysaccharides serve as alternatives that feature antifouling capacity equivalent or even superior to that of PEG[21,22]. Furthermore, recent evidence has revealed that the nonspecific resistance to protein adsorption provided by antifouling materials does not necessarily afford the best therapeutic outcome. In one aspect, the adsorption of some types of proteins, especially those belonging to the dysopsonin family, could remodel the identity of nanomaterials to one invulnerable to immunosurveillance, thereby extending their blood circulation period[23,24]. In addition to this aspect, with regard to feasibility, completely resisting protein adsorption in vivo is difficult[23,25]. Overall, it is now increasingly agreed that controlling the protein corona (i.e., shifting the adsorption of dysopsonins versus opsonins)[26-28], rather than trying to realize its total elimination, would allow the nanomedicine to function better.

In addition to experimental findings[29-37], computational progress has also advanced our understanding of the protein corona, especially at the molecular level[38-41]. In one demonstration, a multi-scale approach was developed to assess the role played by glucose and cholesterol during fibrinogen protein adsorption[38]. Likewise, using dissipative particle dynamics simulation, our group[41] revealed that adsorbed human serum albumin (HSA) protein was implicated in the interaction of nanoparticles with membranes. These computational findings give us insights into the mechanism underlying the interaction of nanoparticles with proteins and, more importantly, show how the protein corona governs the identity of nanoparticles.

In this study, we demonstrate that the combined efforts of experimental and computational investigations can unveil a robust yet simple means of controlling the protein corona. In brief, in vitro analyses and all-atom molecular dynamics (MD) simulations reveal that the availability of hydroxyl groups can affect the interaction of nanomaterials (herein, graphene and gold) with three representative serum proteins (HSA, IgE, and ApoE). Their different adsorption behaviors, in terms of both number and configuration, strongly influence the composition of the subsequently formed protein corona, as confirmed by

proteomics. We thus propose that coating nanomaterials with one type of serum protein before blood entry can help adjust their pharmacokinetics and that ApoE encasement unexpectedly leads to extended blood circulation, as observed for graphene, gold, and iron oxide nanomaterials. Our study not only showcases how theoretical and experimental efforts can benefit from each other but also offers a promising approach to achieve long blood circulation via simple chemistry.

## Results

**Protein adsorption profiles.** We investigated whether and how the surface hydrophilicity of nanomaterials (graphene[42] and gold[43]) influenced their interaction with proteins. In this case, pure graphene (PG) and fully and partly hydroxylated graphene (denoted by G-all-OH and G-half-OH, respectively) were first chosen, and prepared according to published protocols (Fig. 1a, b)[44]. Next, they were incubated with fresh mouse serum. We screened the most abundant proteins in the corona using proteomics. Regardless of the surface functionalization, the lipoprotein and immunoglobin classes accounted for noticeable portions of the corona (Supplementary Fig. 1), which is also in agreement with a previous study[45]. On this basis, we selected ApoE and IgE as representatives of the lipoprotein and immunoglobin families, respectively, given their profound implication in metabolism as well as their great abundance in vivo. Likewise, HSA protein was chosen as a model protein for comparison since it is the most abundant protein present in human blood plasma[46]. Taking G-all-OH as an example, the morphology of nanomaterials coated with different types of proteins was analyzed by atomic force microscopy. In Fig. 1c, HSA molecules passivating the nanomaterials tended to stay isolated from each other, whereas adsorbed ApoE and IgE molecules appeared to be connected and created a thin layer-like corona. The successful formation of coronas consisting of a single type of protein was also confirmed by the increase in hydrodynamic diameter relative to that of pristine G-all-OH (Fig. 1d). To quantify the difference in adsorption, we replaced the proteins with fluorescently labeled proteins while leaving other parameters unchanged. Of note, the introduction of a fluorescent dye into the protein might slightly change the manner of its interaction with substrates but not to a great extent[47]. To avoid misleading results, the comparisons here and thereafter were all made between groups using the same type of proteins, that is, fluorescent or nonfluorescent. The amount of proteins adsorbed was measured based on the fluorescence intensity, as shown in Fig. 1e. In contrast to PG, G-half/all-OH diminished the adsorption of HSA and IgE proteins because of the presence of hydroxyl groups. Of note, we found that the availability of hydroxyl groups was inversely correlated with the adsorption of these two proteins and that this difference was statistically significant in both cases (**$p < 0.01$, $p$ values were calculated using multiple $t$ tests). The ApoE protein tends to interact similarly with all three types of materials, revealing the negligible relevance of the hydroxyl groups in this case. This result was unexpected but ought to help guide the design of graphene-based nanomedicines. Previous evidence revealed that the in vivo adsorption of the ApoE protein, a member of the dysopsonin family[23,48], could prolong the blood availability of endogenous components, and this property is generally referred to as stealth. In contrast, proteins such as HSA and IgE belong to the family of opsonins[24], whose adsorption promotes the clearance of substances. Overall, these findings confirmed that the introduction of hydroxyl groups could assist in reducing the adsorption of those bad proteins negatively affecting the blood availability of nanomaterials while negligibly affecting the passivation of their counterparts serving a protective role.

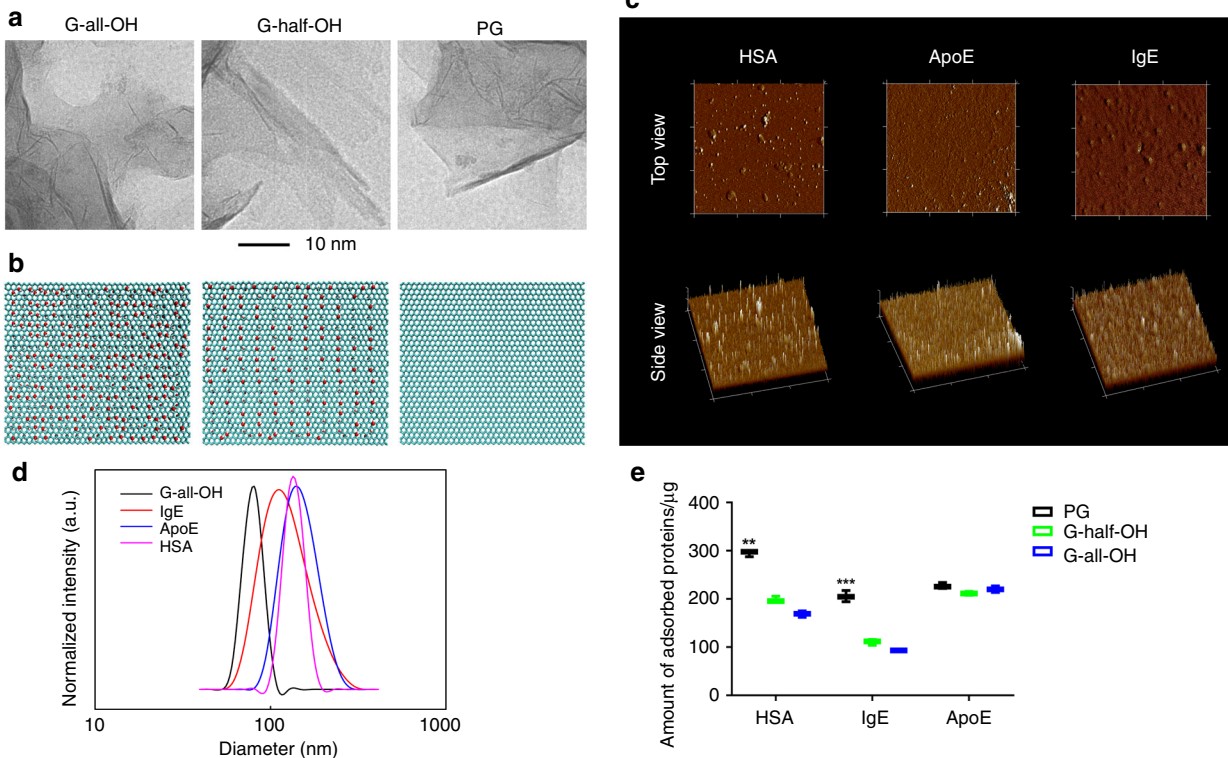

**Fig. 1** Structural information of Graphene and derivatives and their in vitro interaction with proteins. **a**, **b** Graphene sheet (cyan) with different surface modifications in the experiment and simulation. PG represents the pristine graphene, while G-all-OH stands for the graphene with the hydroxyl groups decorated onto both sides of the whole basal plane, and the hydroxyl groups on G-half-OH was roughly half of that exposed by G-all-OH, respectively. **c** AFM results of G-all-OH passivated with HSA, ApoE, or IgE, respectively. **d** Results of DLS showing the size changes post protein adsorption. **e** In vitro analyses showing the number of proteins adsorbed on the graphene. Data are presented as mean ± s.e.m. ($n = 3$). $p$ Values were calculated using multiple $t$ tests (***$p < 0.001$, **$p < 0.01$)

**Molecular modeling of graphene–protein interactions.** Changes in both the net charge and surface chemical properties can be behind differences in attraction to proteins. After analyzing the zeta-potential of G-all/half-OH and their pristine counterpart, we noticed that they all carried slightly negative net charges ($-0.24 \pm 0.1$, $-0.67 \pm 0.2$, and $-0.33 \pm 0.1$ mV for PG, G-all-OH, and G-half-OH, respectively, ± stands for the standard deviation), revealing that a simple electrostatic force-driven interaction with proteins was not the case here. To provide an explanation for these puzzling experimental findings and obtain an in-depth under-standing of the graphene-protein interaction, we applied all-atom MD simulations to investigate the protein adsorption behaviors of graphene sheets subjected to different surface modifications. As shown in Fig. 2a, we initially placed HSA above the sheet in the simulations. After 200 ns (i.e., at the end of the simulation), HSA could adsorb onto the surface of the graphene in all cases. In the case of PG, the adsorption of HSA seemed to be reasonably firm given its many residues contacting the graphene sheet. Likewise, we believed that the interaction of HSA with G-all-OH was weaker than that of HSA with PG according to the lower number of adsorbed residues. To better understand the adsorption ability of HSA in all three cases, the contact surface area (CSA) between the protein and the sheet was calculated. As shown in Fig. 2b, the CSA of HSA on PG was approximately 25.8 nm², much larger than that on G-all-OH (10.8 nm²). When the number of OH groups was halved, an intermediate CSA of ca. 17.9 nm² that laid right between the values of G-all-OH and PG was realized. The above simulation indicated that the CSA inversely correlated with the surface hydrophilicity, which agreed well with our experimental findings.

The interaction energy, including the Lennard–Jones (LJ) and Coulomb interactions was calculated (Fig. 2c). Among the three nanomaterials, the LJ interaction was strongest in PG and weakest in G-all-OH, whereas no Coulomb interaction was confirmed in PG owing to the neutrally charged carbon atoms therein. In addition, no obvious difference was observed between G-all-OH and G-half-OH in terms of the Coulomb interaction. These results suggested that there existed a positive correlation between the CSA and the LJ interaction and were consistent with previous assumptions[49,50].

Furthermore, we assessed the type and number of protein residues adsorbed on graphene (Fig. 2a). A total of 14, 9, and 7 hydrophobic residues were found to be adsorbed on PG, G-half-OH, and G-all-OH, respectively. Of note, the residues determined in each case overlapped to a great extent (see the top view in Fig. 2a). In addition, the number of adsorbed residues of all types, including the hydrophobic, non-charged, and charged hydro-philic residues, was measured to be 34, 26, and 17 for PG, G-half-OH, and G-all-OH, respectively. This order follows the same sequence as did the trend in the number of adsorbed hydrophobic residues. This result drives us to propose that the adsorption of hydrophobic residues of HSA might be linked to the adsorption of residues of other types, probably hydrophilic residues. Normally, for naturally occurring HSA proteins, their hydro-phobic residues are not exposed. As such, if they came into contact with graphene, the proteins should reconfigure to become flatter, which unintentionally promotes the interaction of other residues with graphene, thereby leading to increased adsorption. This supposition was further supported by the fact that increased hydrophilicity disfavored HSA adsorption (Fig. 2b).

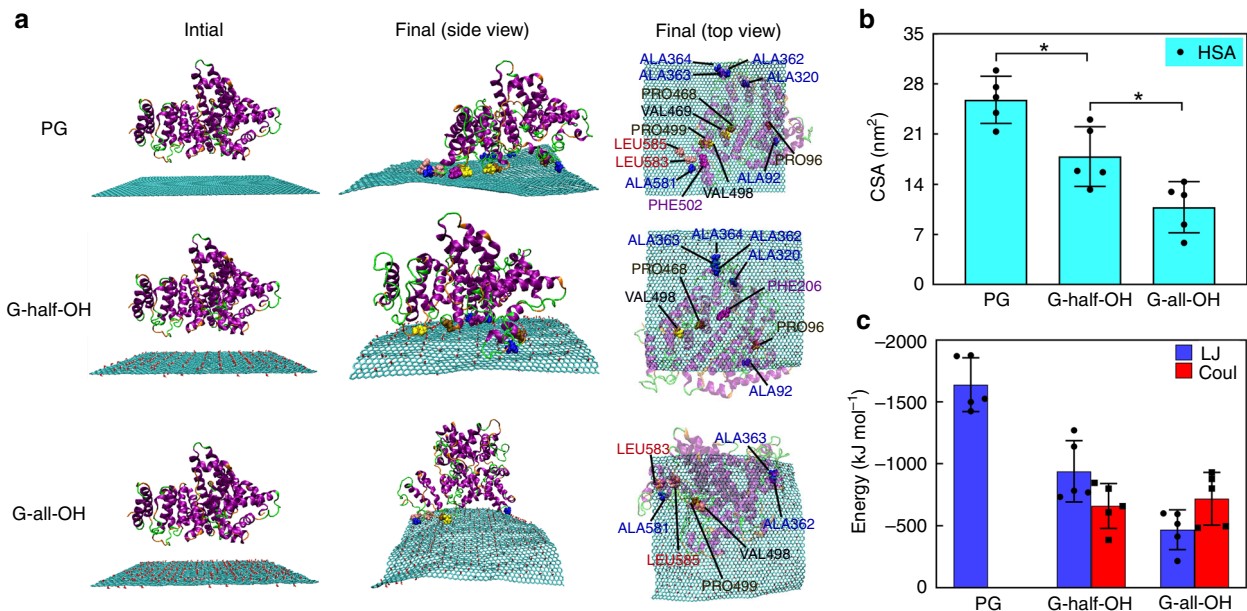

**Fig. 2** The simulation results of the interaction of HSA with the graphene sheet. **a** Typical snapshots of the initial and final structures of the adsorption of HSA onto the sheet (cyan) with different surface modifications. From the side and top views, hydrophobic residues within 0.5 nm distance from the sheet surface are represented as CPK spheres. Water molecules and ions are omitted for clarity. **b**, **c** The contact surface area (CSA) and the Lennard–Jones (LJ) and Coulomb (Coul) interaction energy between HSA and the graphene with different surface modifications, respectively. Data are presented as mean ± s.e.m. ($n = 5$). $p$ Values were calculated using multiple $t$ tests (*$p < 0.05$)

Next, we investigated the adsorption profiles of IgE. As shown in Fig. 3a, IgE adsorption was confirmed in all cases. Based on snapshots from the side view, the variation in protein structure caused by adsorption was seemingly nonsignificant with respect to that observed in the HSA case. Moreover, the CSA results suggested that the adsorption of IgE in these three cases was still different (Fig. 3b). The order of CSA values in this case (16.9, >12.9, > 10.8 nm² for PG, G-half-OH, and G-all-OH, respectively) was the same as that for HSA, possibly owing to their similar hydrophobicity. This result also agrees well with our experimental findings (Fig. 1e), wherein the lowest protein adsorption was observed on G-all-OH. In addition, Fig. 3c shows the LJ interaction and Coulomb interaction between IgE and graphene differed with the surface modifications of graphene. The difference in CSA can be primarily attributed to the distinct LJ interactions between the IgE and the different types of graphene. In addition, we also analyzed the profiles of adsorbed residues (Fig. 3a). The number of hydrophobic residues available on PG was the greatest among all three types of graphene (5 hydrophobic residues), whereas the number of hydrophobic residues for G-half-OH and G-all-OH was 4 and 3, respectively. Similar to HSA, the number of adsorbed hydrophobic residues and the corresponding number of total adsorbed residues (23, 18, and 16 for PG, G-half-OH, and G-all-OH, respectively) followed the same order. The difference from HSA was that IgE had fewer adsorbed hydrophobic residues under identical conditions, which led to a reduced CSA value (Fig. 3b) accompanied by a weakened binding strength.

By employing the same approach, we analyzed the ApoE adsorption profiles. Interestingly, no noticeable difference in adsorption was observed among all three cases, as shown in Fig. 4a, b. This result indicated that the adsorption of ApoE was unaffected by hydrophilicity variations. In particular, an equivalent number of hydrophobic residues could be found adsorbed on PG and G-half-OH. As shown in Fig. 4c, the LJ interaction underwent a minor reduction after the introduction of hydroxyl groups to graphene. The total number of adsorbed residues

remained almost constant, and the CSA value was observed to be independent of hydrophilicity.

Overall, although the adjustment of hydrophilicity by tuning the hydroxyl group availability of graphene can influence the protein adsorption behavior, their manner of action in terms of CSA was affected in different ways. Specifically, for HSA and IgE, their numbers of adsorbed hydrophobic residues and the CSA decreased with increasing number of hydroxyl groups. By contrast, no positive correlation existed between the CSA of ApoE and sheet hydrophilicity.

**Effect of secondary structure change on protein adsorption.** During the adsorption of proteins, their secondary structures could reconfigure or even be compromised due to the strong LJ interaction and/or Coulomb interaction between the nanomaterials and the proteins. Still using graphene as an example, as shown in Fig. 5a, we noticed that the secondary structure of these proteins changed after packing onto graphene (compared with that of the free proteins). In each case, we tried to understand how the change in secondary structure took place. For HSA, we noticed that the larger the helix ratio of the proteins was, the smaller their CSA appeared to be. In other words, the change in the helix ratio (profile of the free protein minus that of its adsorbed counterparts) was positively correlated with the CSA. Similar phenomena were also observed in the case of IgE (helix + β-sheet) and ApoE. To explain this result, we calculated the hydrophobic moments (µH) of HSA and ApoE; µH is a useful parameter for the measurement of helix amphiphilicity[51]. The average µH values of HSA and ApoE were measured to be 0.2876 and 0.3892, respectively (both less than 0.5), indicating that their hydrophobic and hydrophilic residues were distributed homogeneously. As such, it was not favorable for either type of protein to adjust its orientation in order to reconfigure the hydrophobic/hydrophilic domains on the graphene surface. On the other hand, a change in the protein secondary structure (i.e., reduction in the helix ratio) led to the redistribution of hydrophobic and

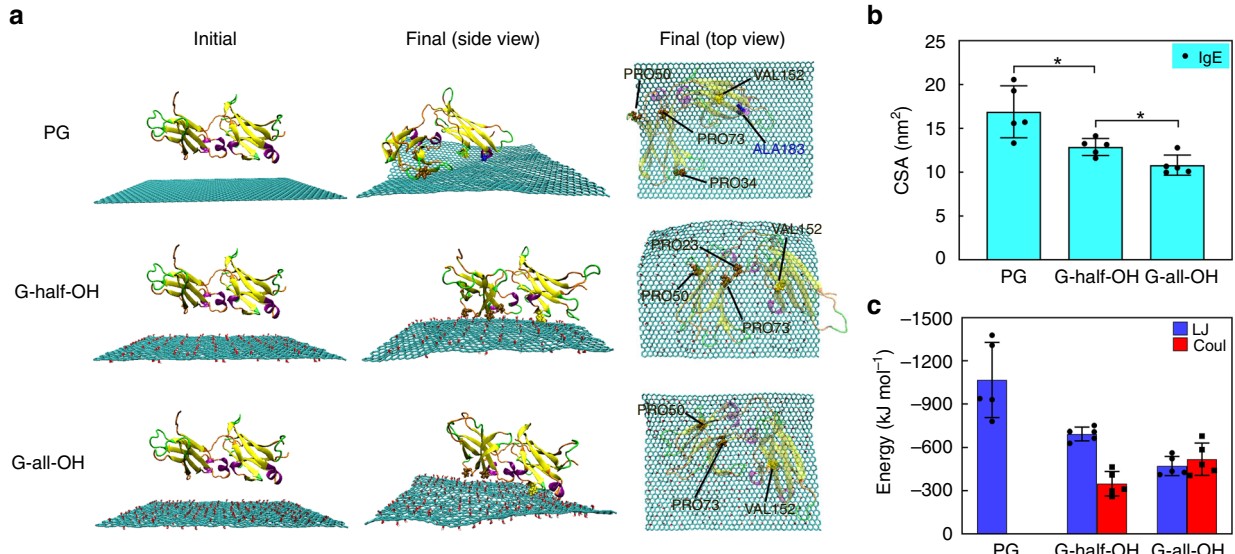

**Fig. 3** The simulation results of the interaction of IgE with the graphene sheet. **a** Typical snapshots of the initial and final structures of the adsorption of IgE onto the sheet (cyan) with different surface modifications. From the side and top views, hydrophobic residues within 0.5 nm distance from the sheet surface are represented as CPK spheres. Water molecules and ions are omitted for clarity. **b**, **c** The contact surface area (CSA) and the Lennard-Jones (LJ) and Coulomb (Coul) interaction energy between IgE and the graphene with different surface modifications, respectively. Data are presented as mean ± s.e.m. ($n = 5$). $p$ Values were calculated using multiple $t$ tests ($*p < 0.05$)

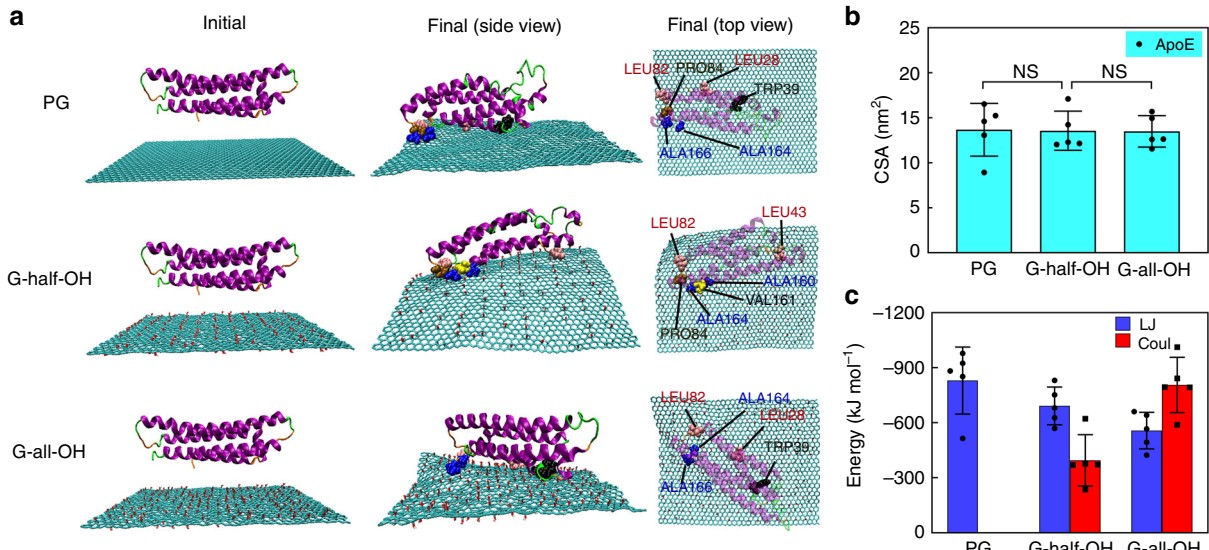

**Fig. 4** The simulation results of the interaction of ApoE with the graphene sheet. **a** Typical snapshots of the initial and final structures of the adsorption of ApoE onto the sheet (cyan) with different surface modifications. From the side and top views, hydrophobic residues within 0.5 nm distance from the sheet surface are represented as CPK spheres. Water molecules and ions are omitted for clarity. **b**, **c** The contact surface area (CSA) and the Lennard–Jones (LJ) and Coulomb (Coul) interaction energy between ApoE and the graphene with different surface modifications, respectively. Data are presented as mean ± s.e.m. ($n = 5$). $p$ Values were calculated using multiple $t$ tests (NS stands for statistically insignificant difference)

hydrophilic residues, which forced a better match between the proteins and graphene accompanied by a greater CSA. In general, changes in the secondary structures of the proteins, especially the transformation from a helix/β-sheet to a coil/turn, were believed to play an essential role in protein adsorption. Notably, since the proteins reconfigured during absorption, it was necessary to assess whether the proteins could function as expected. Taking ApoE as an example, it was found that the flexibility of the protein (i.e., ΔRMSF, which is a key parameter reflecting biological function in MD simulation[52,53]) at the active sites (residues 136–150) changed considerable post-adsorption. Since the ΔRMSF of most of the residues was below zero, the biological function of ApoE in this case was compromised to some extent (Supplementary Fig. 2). Nevertheless, in the case of G-all/half-OH, the number of residues with a positive ΔRMSF was very close to of the number with a negative ΔRMSF, suggesting that the biological function of ApoE was well preserved in these two cases. This result also shows that the presence of hydroxyl groups offers a promising way to minimize the denaturation effect exerted by graphene materials.

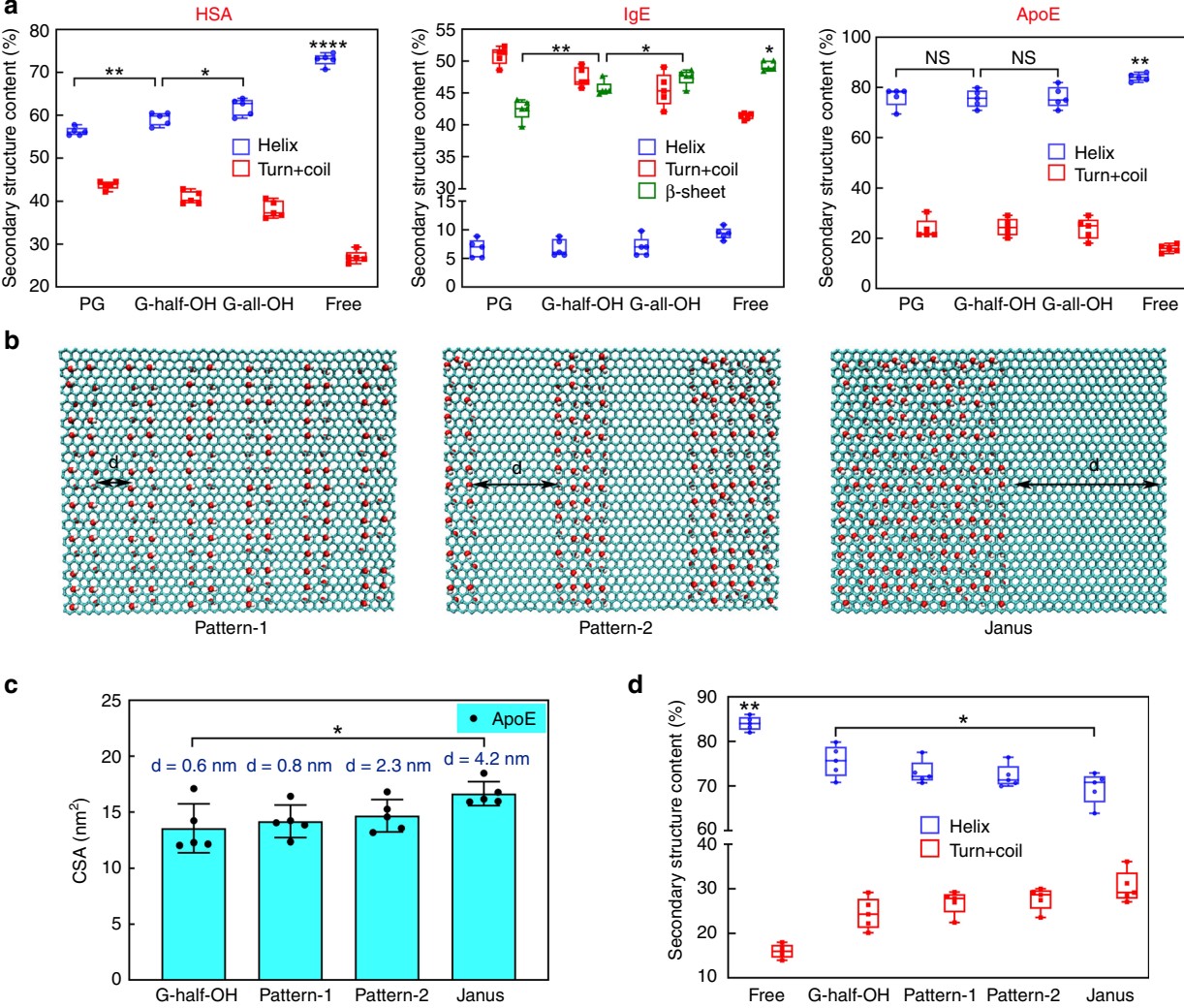

**Fig. 5** Structural profiles of adsorbed proteins. **a** Secondary structure content of different proteins on the graphene after 200 ns MD simulation (Free: the secondary structure content of the proteins in the water). Data are presented as mean ± s.e.m. ($n = 5$). **b** Schematic illustration of three new designed graphene sheets (Janus, pattern-1 and pattern-2) in the simulation: they have the same number of hydroxyl groups as G-half-OH, whereas the modified area (modified with hydroxyl groups) and the unmodified area (pristine graphene) are distributed at intervals, the distance of the interval (defined as d) is about 4.2, 2.3, and 0.8 nm, respectively. **c**, **d** The CSA and the secondary structure content of ApoE adsorbed on the different sheets, respectively. Data are presented as mean ± s.e.m. ($n = 5$). In the box plot, the upper and lower quartile as outlined by top and bottom boundary were divided by line showing the median value. $p$ Values were calculated using multiple $t$ tests (****$p < 0.0005$, **$p < 0.01$, *$p < 0.05$, and NS stands for statistically insignificant difference)

**Effect of hydroxyl group distributions on protein adsorption.** Next, we progressed slightly further by assessing whether and how the surface pattern of hydroxyl groups affects the graphene–protein interaction. In this case, we focused on ApoE and investigated its adsorption onto different graphene sheets bearing three proposed patterns (Fig. 5b). Specifically, the distance between neighboring hydroxyl groups in the three patterns, denoted by Janus, pattern-1, and pattern-2, was set to 4.2, 2.3, and 0.8 nm, respectively, whereas the distance in G-half-OH was approximately 0.6 nm. First, we analyzed the CSA value. The calculated CSA was distance-dependent, and the greater the distance between neighboring hydroxyl groups was, the greater the CSA appeared to be (Fig. 5c). We speculated that the difference in the helix ratio change of the four different classes of graphene sheets brought about this observation. As shown in Fig. 5d, the helix ratio declined when the distance increased, indicating that the "fitting" potential of the ApoE protein could be augmented when there existed a clear boundary separating the hydrophobic and hydrophilic domains. Notably, this result agrees well with our

previous findings, wherein the Janus distribution of two ligands on nanoparticle surfaces favored their internalization into cells carrying the corresponding receptors[54].

As mentioned above, the ApoE protein could rearrange to somewhat fit the hydrophobic and hydrophilic domains on the graphene surface, although complete reconfiguration is not feasible. It is thus reasonable to see some bad adsorption in certain cases. In addition, there might also exist some unexpected interactions between the residues and the nearby domains of the nanomaterials (in this case, the hydrophobic residues cannot just adsorb onto the hydrophobic region and may also interact with the hydrophilic region). Overall, the limited distance between the regions of the hydroxyl groups could hinder protein adsorption, thereby leading to a smaller CSA. To maximize ApoE adsorption, graphene carrying Janus-like hydroxyl groups might be the best choice.

**Protein adsorption profiles of Au.** We sought to determine whether the tuning of protein adsorption through modification of

surface hydroxyl groups can also be extended to materials other than graphene. Based on the considerations of biocompatibility and potential as nanomedicines, we selected gold (Au) as a material[43]. To precisely control the number of surface hydroxyl groups, we followed a recent protocol[55], wherein a complete ligand exchange on Au nanomaterials was allowed with the assistance of silver deposition. Briefly, we synthesized 50-nm Au nanospheres in an aqueous system, followed by the deposition of a thin layer of silver (Ag) to help extrude the original capping agents (schematic illustrated in Fig. 6a). Next, the intermediate Au@Ag nanospheres were exposed to a solution containing $H_2O_2$ and an alkylthiol/thiol-terminated alkyl alcohol (C8), which sequentially allowed the removal of the Ag layer and thiolation. As no direct ligand exchange is required here, the desired molecules can passivate the Au nanospheres stoichiometrically, making it feasible to create Au nanospheres that differ in surface functionalization by simply tuning the feeding amount of the ligands. We analyzed the protein adsorption profiles of Au nanospheres under conditions identical to those used for studying graphene. As shown in Fig. 6b, the introduction of hydroxyl groups led to a marked reduction in the attraction of Au to both the HSA and IgE proteins. Of note, there was no statistically significant difference between the fully and partially hydroxylated Au nanospheres in terms of the amount of adsorbed proteins. Nevertheless, the hydroxyl groups did not affect the interaction between the nanomaterials and ApoE proteins. Further, we assessed the protein configuration using MD simulation and analyzed the CSA and LJ/Coulomb interaction profiles. Figure 6c and d shows the snapshots collected at the end of simulation and the corresponding CSA data, respectively (for a full spectrum of images, please refer to Supplementary Figs. 3–5). Consistent with the trend in the case of graphene, there also exists a trend wherein decreased CSA can be observed in the HSA and IgE cases but not in the ApoE case. Moreover, we noticed that the changes in the number of hydrophobic residues available on the Au nanospheres, as well as the interaction energy, followed the same trend that CSA did. Altogether, these findings strongly supported the notion that hydrophilicity—in other words, the presence of hydroxyl groups—helps remodel how nanomaterials interact with adsorbed proteins. Regarding the protein adsorption behavior, encasement of hydroxyl groups might influence the substrate differently depending on the materials chosen as the substrate. In addition, we speculated that the size, morphology, and even curvature of the engineered nanomaterials might affect the manner of hydroxyl group–protein interaction to some extent. It is thus difficult to compare the difference in the absolute number of proteins among distinct nanomaterials featuring similar surface functionalization. Furthermore, for a given type of material, at least for Au and graphene, the correlation of reduced passivation by HSA and IgE but no obviously reduced passivation by ApoE with the presence of hydroxyl groups was established and comprehensively validated.

**In vivo biodistribution profiles of nanomaterials**. We next tested the idea of pre-adsorbing either IgE or ApoE (HSA was excluded due to the compatibility concern) onto nanomaterials in the hope of changing their pharmacokinetics. To assess their biodistribution, G-half/all-OH were first modified and chelated with Gd(III) to make them measurable by inductively coupled plasma-mass spectrometry (ICP-MS). As we have validated in vitro, the Gd-tag remained almost intact (defined as less than 2% loss in amount) for at least 3 days (Supplementary Fig. 6), meaning that the measurement of Gd content in vivo faithfully reflected the amount of graphene. The Gd-labeled G-half/all-OH samples were passivated with IgE/ApoE before injection into

tumor-bearing mice through the tail vein. Uncoated G-half/all-OH samples were assessed under identical conditions for comparison. We sacrificed these mice at various time points after injection and collected several organs and blood samples.

Figure 7a shows the blood availability of different types of G-half/all-OH as a function of time. Of note, the pristine ones and those coated with IgE experienced fast washout from blood circulation, with both having a half-life shorter than 4 h. This result agrees well with the substantial increase in accumulation in mononuclear system-related tissues such as the liver and spleen. Regardless of the hydroxyl group availability, pre-adsorption with ApoE can result in markedly extended blood circulation. Previous findings revealed that passivation with either IgE or ApoE creates a thin layer of an intact protein-based shell (Fig. 1d). We envisioned that this type of corona, rather than the nanomaterial itself, regulates the interaction with additional serum proteins that determines the fate of the whole substance. To verify this hypothesis, we replaced graphene with Au nanospheres, and the bio-distribution analysis was repeated without changing any other parameters. As Fig. 7b shows, the results, in which pre-coating with ApoE protein favored a prolonged half-life of Au nanospheres that differed in surface functionalization, can be replicated. In both cases, as shown in Fig. 7a, b, the longer these nanomaterials can circulate in blood, the more they can be passively enriched in tumor tissue, presumably owing to the enhanced permeabilization and retention effect.

We also extended this approach to iron oxide, a substance well known for its ability to enhance the $T_2$ contrast under magnetic resonance imaging. As the images show (Supplementary Fig. 7), obviously reduced liver accumulation can be observed when the nanoparticles were coated with ApoE prior to the administration compared with their IgE-coated or pristine counterparts, in which cases the liver tissue was darker.

**Mechanism underlying the action of pre-coating with proteins**. We next tried to understand how pre-adsorbed IgE and ApoE proteins function to modulate the bio-distribution of nanomaterials. Using graphene as the substrate, we evaluated the composition of the protein corona using proteomic analysis. Some may argue that either IgE or ApoE might stand as the dominant species being recognized because of pre-adsorption. To avoid misleading results, a chart showing the population of various types of proteins was generated based on the number, rather than the abundance, of corresponding proteins (Fig. 8a). We noticed that previous passivation with ApoE contributed to resistance to complementary proteins and immunoglobins, whereas IgE had exactly the opposite effect. This observation is once again independent of the surface properties of the substrate, revealing that it is likely an effect exerted by the pre-adsorbed proteins. The uptake profiles of macrophages (RAW 264.7 cells) further confirmed that the ApoE coating rendered the materials less vulnerable to macrophage recognition and consequent engulfment (Fig. 8b). This result can be partly explained by the limited participation of the complementary system herein, which further rationalized the enhanced cellular uptake of IgE-coated graphene compared to PG. When graphene was replaced by Au nanospheres, a similar outcome was realized, as reflected by the results of ICP-MS analysis (Fig. 8c). This evidence collectively suggests that ApoE plays a role as a dysopsonin that counterbalances the biofouling effect associated with opsonins, the role of which is played by IgE. Pre-coating nanomaterials with both types of proteins likely recruits a cascade of proteins that differ in types and functions, eventually leading to distinct coronas that differently govern the fate of nanomaterials.

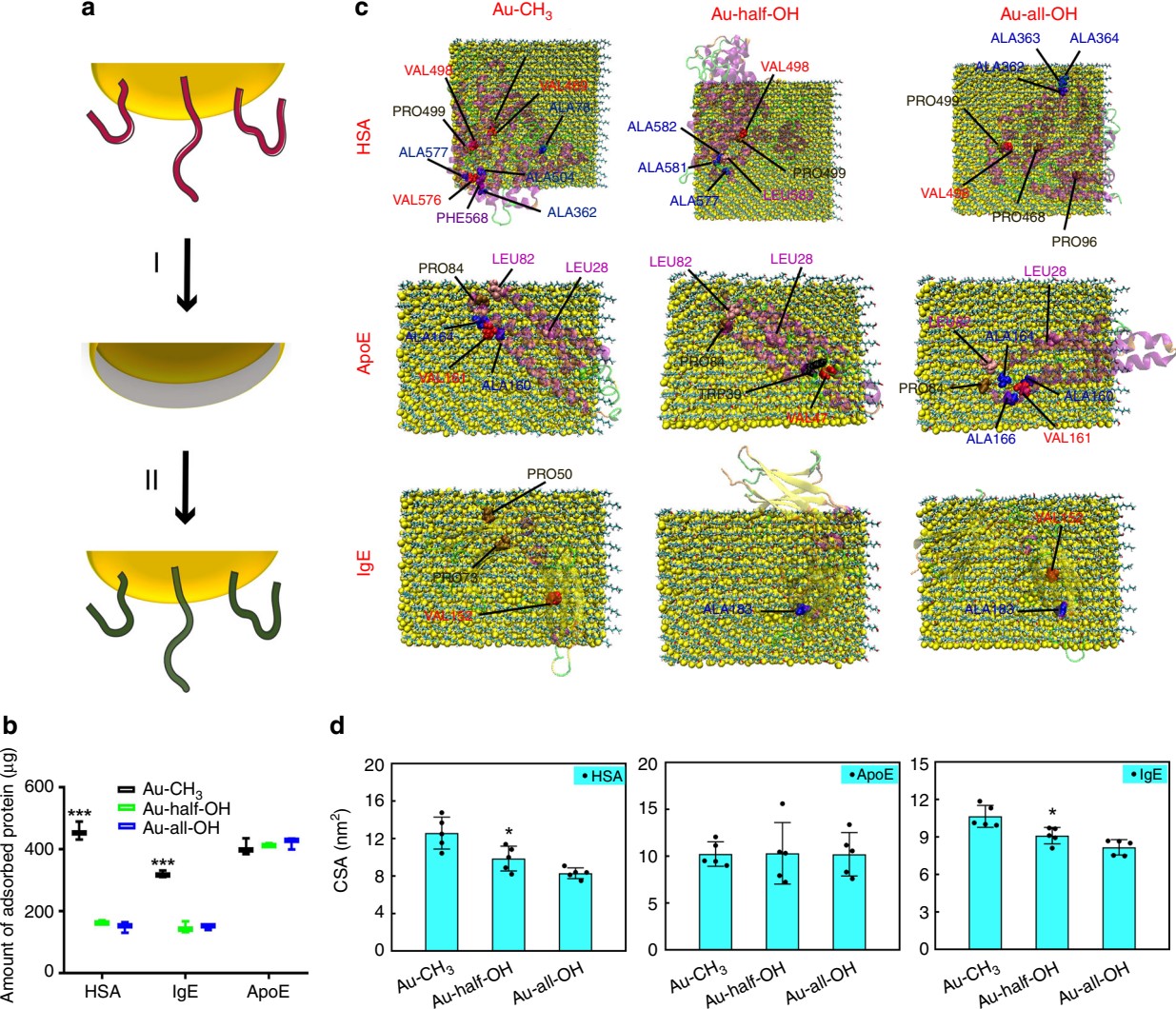

**Fig. 6** Profiles of protein adsorbed on Au nanoparticles. **a** Schematic illustrating the step-wise introduction of foreign ligands with the assistance of Ag. I: Deposition of a thin layer of Ag; II: Removal of Ag alongside with the passivation with ligands. **b** In vitro analyses showing the number of proteins adsorbed on the graphene. Data are presented as mean ± s.e.m. ($n = 3$). $p$ Values were calculated using multiple $t$ tests (***$p < 0.001$). **c** Typical final snapshots of the adsorption of three types of proteins onto the Au nanoparticle (yellow) with different surface modifications. The hydrophobic residues within 0.5 nm distance from the surface are represented as CPK spheres. Water molecules and ions are omitted for clarity. **d** The CSA profiles of proteins adsorbed on different Au nanospheres. Data are presented as mean ± s.e.m. ($n = 5$). $p$ Values were calculated using multiple $t$ tests (*$p < 0.05$). For a full spectrum of snapshots and interaction energy profiles, please refer to Supplementary Figs. 3–5

**Association of pre-coated proteins with tissue accumulation**. A previous study demonstrated that some ligands might not stay conjugated to nanomaterials after extended blood exposure[56]. We envision that the same findings might be applicable to coated proteins. As such, we employed fluorescently labeled proteins. If the protein corona remains intact, the ratio of the fluorescent intensity of enriched nanomaterials to the Gd content would remain constant; otherwise, any decrease in value would accurately reflect the loss of proteins. Interestingly, we found that those injected nanomaterials featured protein coatings that were compromised to different tissue-dependent extents. Specifically, better preservation of the ApoE coating can be found in the nanospheres that are still circulating in the blood and that accumulate in the tumor (Fig. 9). Their counterparts captured by MPS tissue were all subjected to great loss of the ApoE coating. Upon replacing ApoE with IgE, we found the trend was reversed: the greater the amount of IgE remaining was, the faster the blood clearance experienced by the nanomaterials. In mice bearing

patient-derived xenografts (PDXs) that resemble the pathological features of human cancer (Fig. 9), we replicated this result, showing that the correlation between the protein corona and preference in tissue accumulation was not cancer type-dependent. It remains elusive how nanomaterials carrying roughly the same protein corona experienced substantially different changes in vivo. One possible explanation for this behavior is that exposure to a high-pressure fluid (herein, blood) somehow exaggerates the previously tiny difference between nanomaterials in terms of ApoE availability. This explanation was partly supported by the observation that high tumoural interstitial pressure led to additional loss of adsorbed proteins (Fig. 9), regardless of their type, relative to either the H22 or PDX case, supporting the notion that high pressure played a role in remodeling the protein corona.

**Long-term safety**. We assessed whether pre-coating with ApoE/IgE would have some long-term implications. Graphene was ruled out here due to the significantly weakened stability of Gd

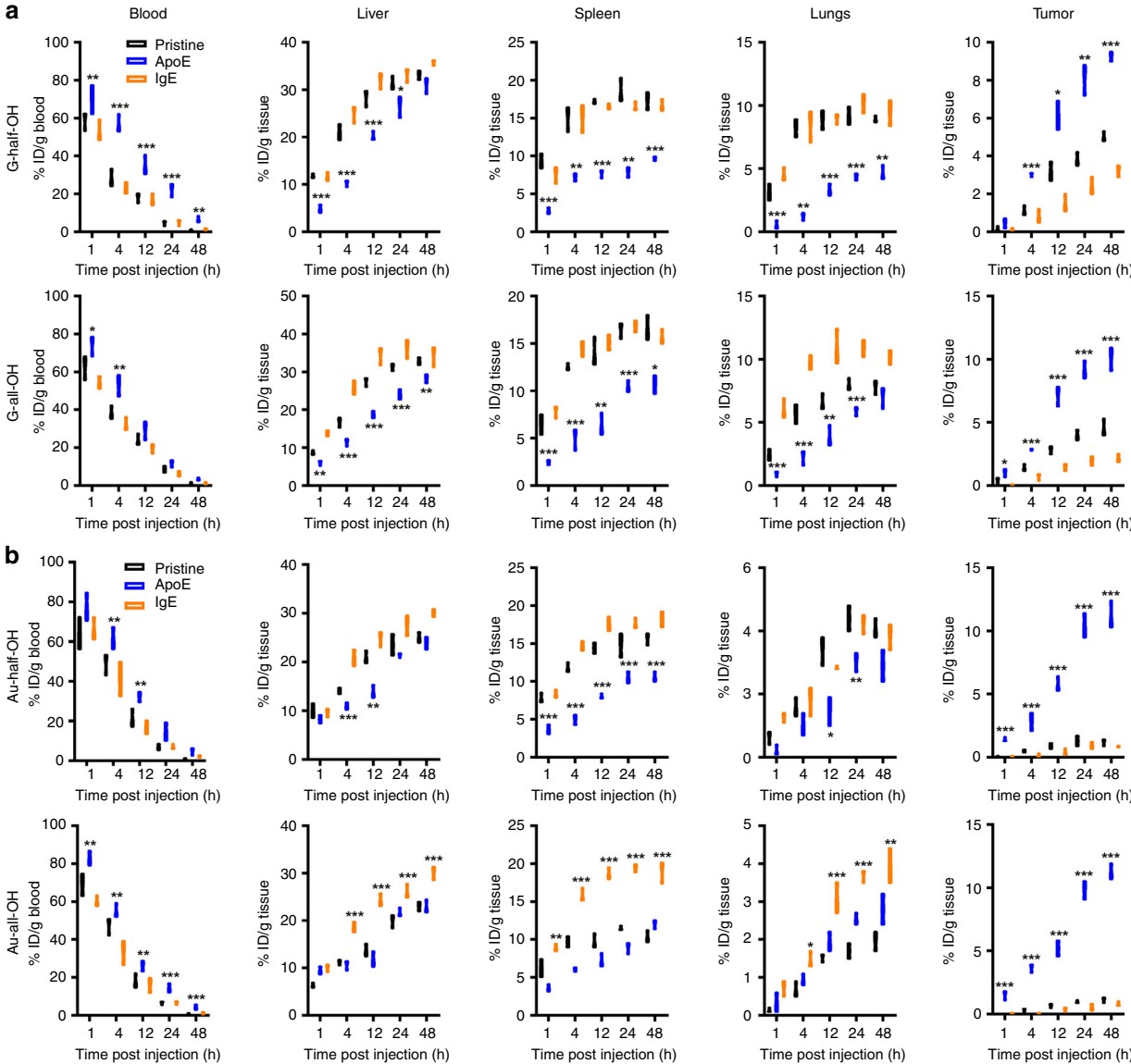

**Fig. 7** Biodistribution profiles. **a**, **b** Biodistribution of G-half/all-OH and Au-half/all-OH in four tissues including liver, spleen, lungs, and kidneys in addition to blood, respectively. Data are presented as mean ± s.e.m. ($n = 3$ at each time point) as a function of time. $p$ Values were calculated using multiple $t$ tests (***$p < 0.001$, **$p < 0.01$, *$p < 0.05$)

labeling beyond one week, so Au nanospheres were selected as the substrate. After 30 days, the injected Au nanospheres, even those that differ in coated proteins, exhibited similar wash-out behaviors (Fig. 10a). Of note, Au nanospheres bearing halved hydroxyl groups tended to stay longer than their completely hydroxylated counterparts, which agrees with previous findings that lower hydrophilicity favors retention in the liver[57]. Collectively considering the abovementioned results, we envision that a transient loss of coated proteins due to a sophisticated niche such as that in the liver and tumor eventually exposes the nanomaterial entirely to the tissue, the clearance from which is thus dependent only on the physiochemical features of the materials. We further confirmed that such a strategy utilizing protein pre-coating was free of chronic complications, as revealed by both pathological assays (Fig. 10b) and hematological studies (Fig. 10c). Notably, the temporal elevation in expression of some biomarkers can be a natural response to foreign species that does not necessarily signify acute inflammation caused by ApoE[58].

## Discussion

Taking several types of broadly employed materials as examples, we showed here that the availability of hydroxyl groups can be harnessed to manipulate the protein corona and contributes a simple yet robust approach to extend the blood circulation of nanomaterials. Combining the efforts of in vitro analysis and computation, we found that increased hydrophilicity substantially compromises the passivation of graphene by HSA and IgE, two members of the opsonin family, whose adsorption results in faster clearance from blood. By contrast, a critical member of the dysopsonin family that plays a role opposite to the opsonins, ApoE, is unaffected by changes in hydrophilicity. That is, a shift in surface properties towards hydrophilicity can effectively help resist passivation by bad proteins while leaving good ones nearly unaffected. We further comprehensively investigated how precoating with proteins remodeled the manner in which nanomaterials are recognized by the MPS and professional phagocytes by using proteomics and fluorescence imaging. All these findings pointed out that prior coating with ApoE helps maintain a corona

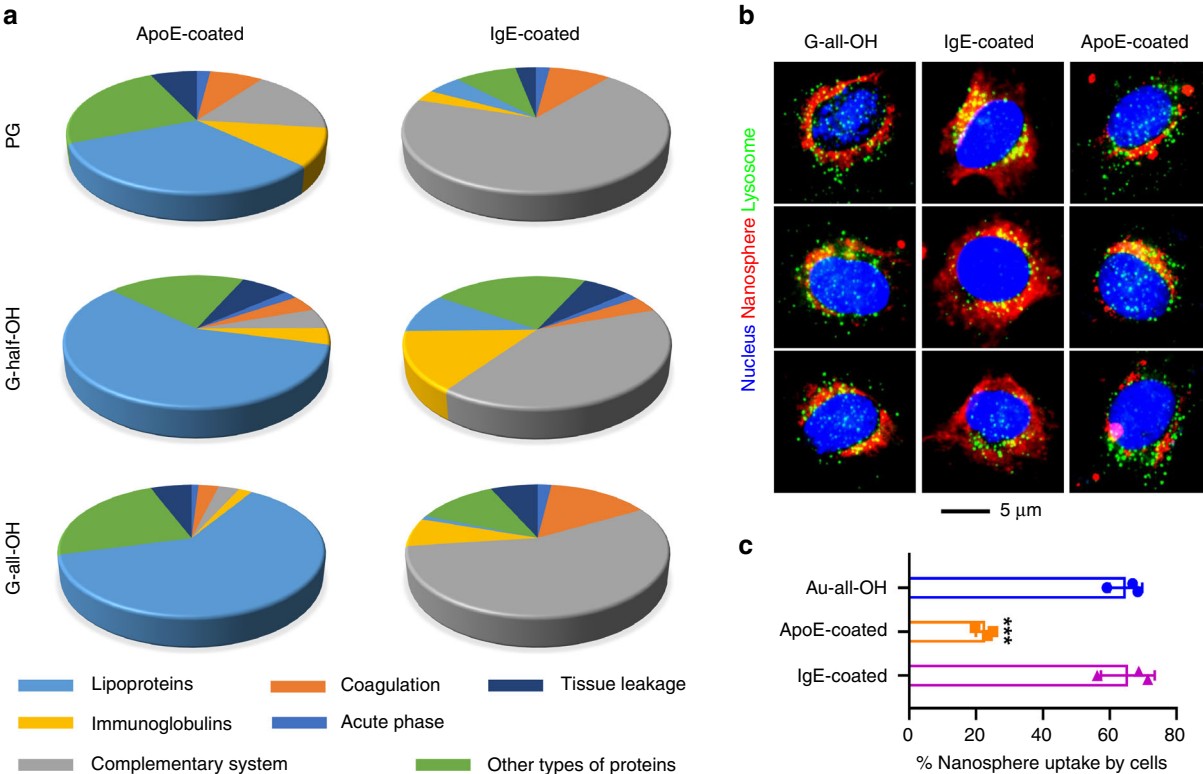

**Fig. 8** Biological implications of pre-adsorbed proteins. **a** Proteomic analysis showing the composition of protein corona encasing G-half/all-OH and PG pre-coated with ApoE/IgE. Of note, the pie-chart was generated based on the number of recognized proteins belonging to each of the seven categories including lipoproteins, coagulation, tissue leakage, immunoglobulins, acute phase, complementary system, and other types. **b** Macrophage uptake of dye-labeled G-all-OH and counterparts coated with ApoE or IgE. Lysosome compartment and nucleus were stained to guide the eyes. **c** Quantified uptake by macrophages of Au-all-OH and protein-coated counterparts. Data are presented as mean ± s.e.m. (n = 3). p Values were calculated using multiple t tests (***p < 0.001)

rich in dysopsonins, which eventually endow the nanomaterials with enhanced enrichment in tumor tissue. Such an effect exerted by ApoE is not long lasting, possibly owing to the gradual loss of attachment during biological fluid exposure, which unintentionally addresses the concern of long-term safety. We believe that this insight into protein–nanomaterials interactions and their association with hydrophilicity, along with the protein pre-coating methods proposed herein, can promote better utilization of nanomaterials' therapeutic power.

## Methods

**Materials.** Ethanol, hydrazine hydrate ($N_2H_4$), sodium bicarbonate ($Na_2CO_3$), sodium nitrate ($NaNO_3$), hydrogen peroxide ($H_2O_2$), potassium permanganate ($KMnO_4$), sulfuric acid ($H_2SO_4$), nitric acid, hydrochloride acid (HCl), ferrous chloride tetrahydrate ($FeCl_2·4H_2O$), and ferric chloride hexahydrate ($FeCl_3·6H_2O$) were all purchased from Sinopharm Chemical Reagent Co., Ltd. (China). Silver nitrate ($AgNO_3$), poly(vinylpyrrolidone) (PVP, MW ~55 kDa), cetyl-trimethylammonium chloride (CTAC), ascorbic acid (AA), gold(III) chloride trihydrate ($HAuCl_4·3H_2O$), sodium borohydride ($NaBH_4$), diethylenetriamine-pentaacetic acid (DTPA) N-(3-dimethylaminopropyl)-N′-ethylcarbodiimide hydrochloride (EDC), N-hydroxy-succinamide (NHS), hydroiodic acid (HI), and gadolinium chloride ($GdCl_3$) were all purchased from Sigma-Aldrich (United States). Natural graphite was obtained from XFNANO Co., Ltd. (China). All chemicals except $N_2H_4$ were of analytical grade and used as received without further purification. Deionized (DI) water with a resistivity of 18.2 MΩ cm was used in all experiments, which was produced by a Millipore ultrapure water system (United States).

**Cell lines.** Mouse breast cancer cell line (EMT-6) and Mouse monocyte/macrophage cell line (RAW 264.7) were all obtained from American Type Culture Collection (ATCC). They were both cultured and maintained in the growth medium of DMEM supplemented with 10% FBS and 1% penicillin/streptomycin. Cultures were maintained in an incubator at 37 °C in a humidified atmosphere of 5% $CO_2$. The medium was replaced every other day until a ca. 90% confluency had

reached. Mouse hepatoma cell line (H22) was obtained from Chinese Academy of Sciences. As a murine ascites cancer cell line, H22 cells were live cultured in mice, and passed regularly to another one once the number of cells reached ca. $5 × 10^8$ cells per mouse.

**Characterization.** The morphology of nanoparticles was investigated by high resolution transmission electron microscopy (TEM-2100, JEOL). The Zeta-potential and DLS profiles of nanomaterials were analyzed at room temperature by using Litesizer TM500 (Anton-Paar). Analysis with atomic force microscopy was conducted on Dimension Icon (Bruker). The proteomic analysis was conducted on hybrid dual-cell quadrupole linear ion trap-Orbitrap mass spectrometer (LTQ Orbitrap Elite, Thermo Fisher Scientific). Fluorescence images were collected on inverted fluorescence microscope (IX83, Olympus) equipped with a live-cell culture chamber (ibidi) supplemented with 5% $CO_2$.

**Animal tests.** Female BALB/c mice (4–6 weeks) and female BALB/c nude mice (6–8 weeks) were both purchased from the Comparative Medicine Centre of Yangzhou University and used for the in vivo tests. All the protocols for the animal tests have been reviewed and approved by the Committee on Animals at Nanjing University and performed in accordance with the guidelines provided by the National Institute of Animal Care.

**Synthesis of nanomaterials.** The G-all-OH were prepared using the Hummers and Offenman's method[59]. The as-prepared G-all-OH was reduced partially[44] to halve the hydroxyl groups, namely G-half-OH. Both G-all-OH and G-half-OH can be dispersed in PBS and were stored at room temperature before future use.

We synthesized 50-nm Au nanospheres by using seed-mediated aqueous solution reduction and corresponding Ag-assisted ligand replacement[55]. Those freshly prepared Au nanospheres carried CTAC capping layer were next washed for two times with DI water, followed by transferring into solution containing PVP (3 mM) and AA (40 mM). After gentle shaking to mix well of these components, aqueous solution containing $AgNO_3$ was introduced. The rapid change in color of the solution indicated the success of Ag deposition, along with the exclusion of previous CTAC ligands. The reaction was allowed to proceed for 10 min before the addition of acetone to lower the solubility of PVP-capped Au@Ag nanospheres.

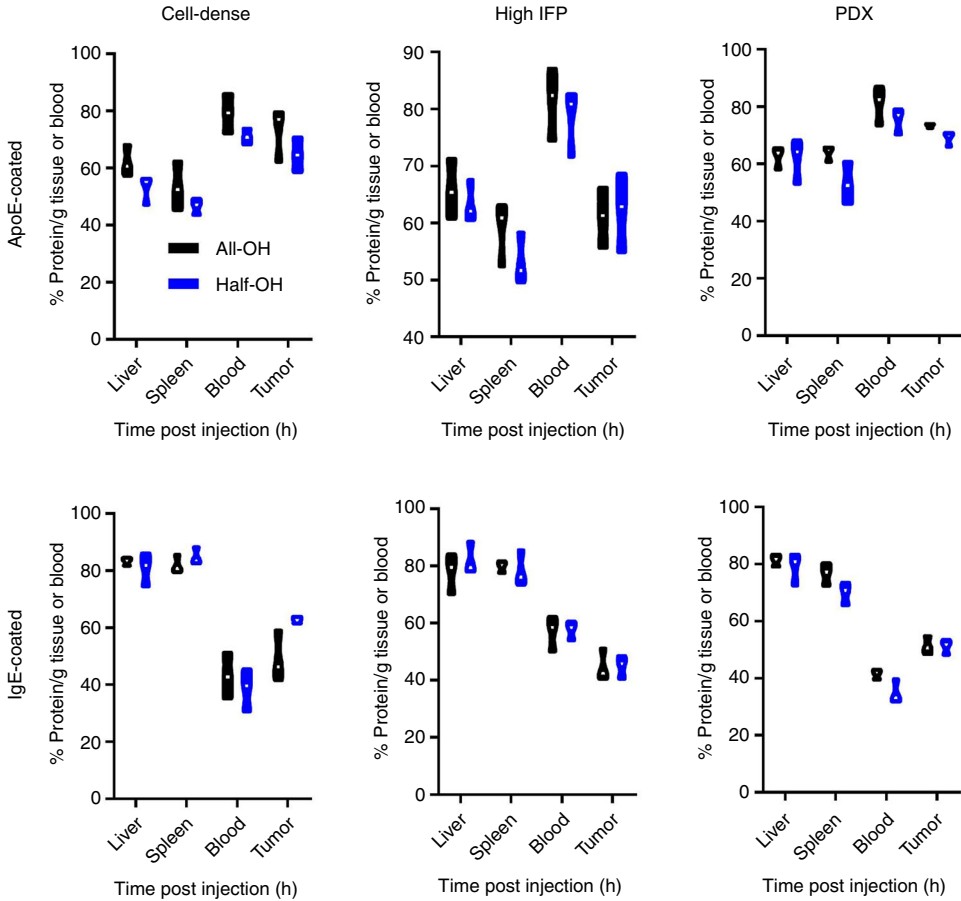

**Fig. 9** Loss of original coating proteins. The amount of pre-adsorbed proteins on nanomaterials were quantified using ICP-MS at 24 h post the administration of G-all/half-OH, with the percentage of remainder capping layer calculated as described in the experimental section. Cell-dense tumor and high IFP stand for the xenograft grown from H22 cells and EMT-6 cells, respectively. The PDX model was established by transplanting the tissue sections harvested from breast cancer-suffering patients into nude mice. Data are presented as mean ± s.e.m. ($n = 3$). In the box plot, the upper and lower quartile as outlined by top and bottom boundary were divided by line showing the median value

We harvested these products through centrifugation and resuspended them in DMF containing either thiol-terminated alkyl alcohol (C8) or alkyl thiol (C8) to create Au-all/half-OH or Au-CH$_3$, respectively. Removal of the Ag layer was induced by the introduction of H$_2$O$_2$, during which the C8 ligands got access to and thereby passivated the outer surface of pristine Au nanospheres. Of note, the mole ratio of C8 ligands to Au nanospheres to prepare fully hydroxylated particles was set to be 1500: 1, whereas the reduced input of C8 ligands at a ratio of roughly 750: 1 against particles was employed to synthesize Au nanospheres bearing halved hydroxyl groups. All these Au nanomaterial derivatives were collected via centrifugation (22,000 $g$ × 30 min) followed by three washes with DI water. They were suspended in phosphate-buffered saline (PBS) and stored at 4 °C and kept from direct sunlight irradiation.

**In vitro analysis of protein adsorption**. All three types of proteins were obtained from Sigma-Aldrich in the form of powder. Before incubation with nanomaterials, they were fluorescently labeled with Alexa Fluor-633 using the peptide labeling kit (Thermo Fisher Scientific, United States). Next, the concentrations of proteins were quantified using BCA assay (Thermo Fisher Scientific), with its correlation with the fluorescence intensity (measured by fluorescence spectrometer, RF-5301PC, Shimadzu) was established to guide the estimation of proteins adsorbed onto nanomaterials. In a typical test, 1 mg of protein was introduced into PBS and mixed well under vigorous shaking before the addition of graphene (final concentration: 10 mg mL$^{-1}$). Afterward, the protein adsorption was allowed to proceed for 4 h under gentle shaking at 37 °C. During this process, the mixture was protected from direct exposure to light. We collected the supernatant of the solution after centrifugation (20,000 $g$ × 30 min), with its fluorescence intensity measured following procedures mentioned above. The number of proteins adsorbed onto nanomaterials were calculated based on the calculated reduction of fluorescence intensity. For the assessment of adsorption onto Au nanospheres, the tests were conducted under identical condition except that the final concentration of nanospheres incubated with proteins was set to be 10$^{14}$ particles per mL.

**All-atom MD simulation**. Similar to the in vitro experiments, three types of graphene sheets (PG, G-half-OH, and G-all-OH), three types of Au nanoparticles coated by alkanethiol chains (Au-CH$_3$, Au-half-OH, Au-all-OH) and three representative serum proteins (ApoE, IgE, and HSA) were considered in the all-atom MD simulations. The starting 3D structures of ApoE, IgE, and HSA were obtained from the Protein Data Bank (PDB) with PDB entry codes 1LE2[60], 4GRG[61], and 1AO6[62], respectively. For HSA, some missing residues were added based on the internal coordinate definitions of the OPLS-AA topology files; while for IgE, for the sake of simplicity, only one chain was employed in the simulations. The pristine graphene (PG) sheets were generated by the VMD software package with each atom modeled as a neutral sp2 carbon. G-all-OH and G-half-OH were the graphene sheets with the hydroxyl groups randomly decorated on both sides of the whole basal plane, where the total number of hydroxyl groups on G-half-OH was roughly half of that on G-all-OH. While for the Au nanoparticles, since its size in the experiments was about 50 nm, much larger than the size of the proteins, here for the sake of simplicity, the flat Au surface was used to model the Au nanoparticle, which was also generated by the VMD software package. Moreover, two types of chains (i.e., the hydrophobic SC chain S-(CH$_2$)$_7$-CH$_3$ and the hydrophilic SO chain S-(CH$_2$)$_7$-CH$_2$OH) was decorated to the Au surface. Au-CH$_3$ and Au-all-OH represented the Au surfaces coated by SC chains and SO chains, respectively; while Au-half-OH was the AU surface simultaneously coated by the same number of SC and SO chains. Due to the different sizes among the three proteins, we chose the size of the graphene sheets with 8.5 × 6.5 nm$^2$ for ApoE and IgE, and 9.8 × 9.2 nm$^2$ for HSA[39], and the size of the Au surface with 8.0 × 6.0 nm$^3$ for ApoE and IgE, and 9.2 × 9.2 nm$^2$ for HSA. As a result, the size of simulation box was 10.5 × 8.5 × 7.0 nm$^3$ for ApoE/IgE-graphene system, 12 × 11.5 × 10.0 nm$^3$ for HSA-graphene system, 10 × 8 × 8.5 nm$^3$ for ApoE/IgE-Au system and 11 × 10.5 × 11 nm$^3$ for HSA-Au system. Initially, the protein and the nanoparticles were placed in a cubic box and placed at a minimum distance of 1.0 nm between any heavy atom of protein and the sheet surface. Then the simulation box was solvated with plenty of water molecules, and finally, Na$^+$ or Cl$^-$ ions were added to neutralize the systems.

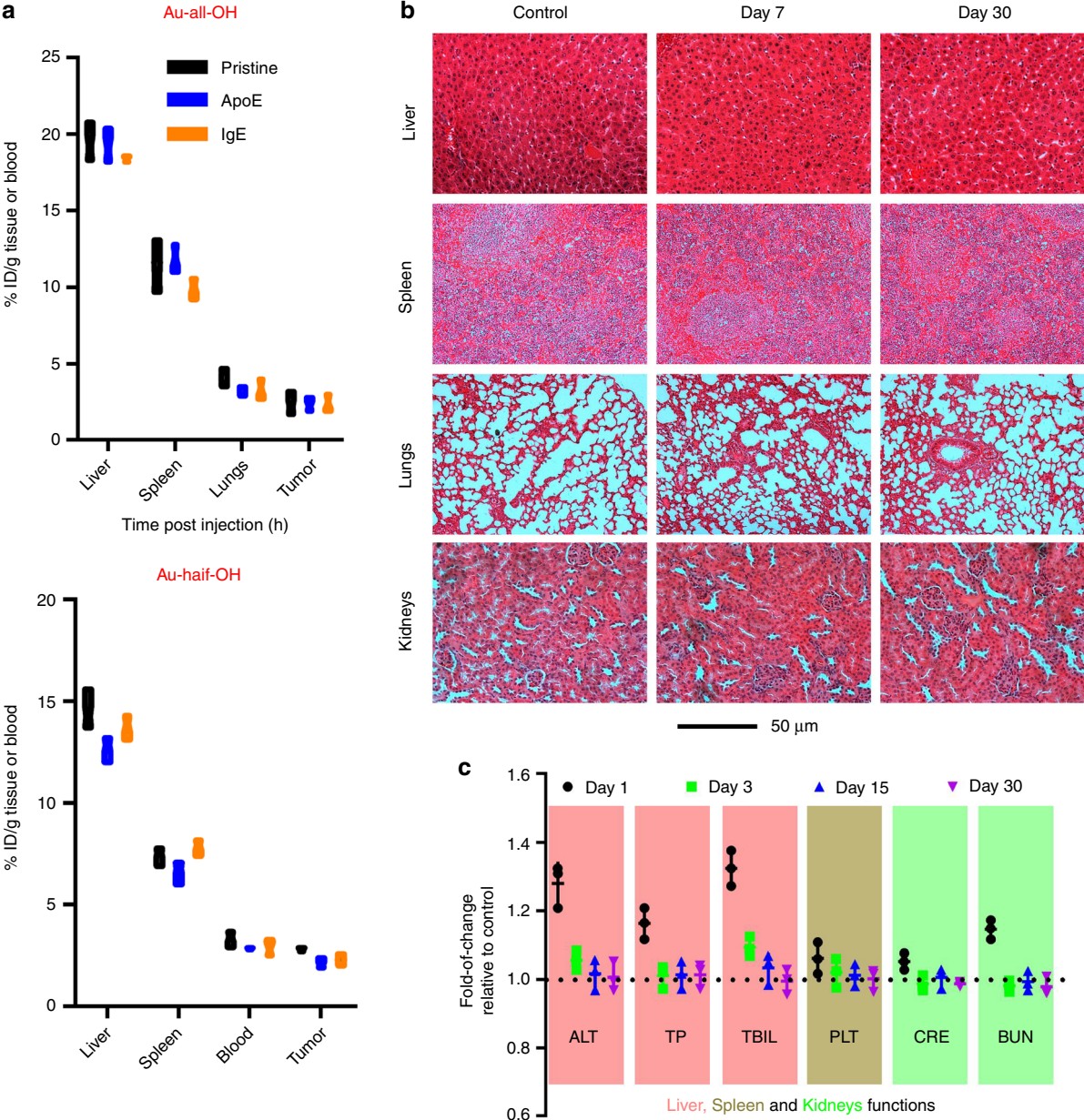

**Fig. 10** Long-term toxicity. **a** Content of Au element remained in blood and three tissues, including liver, spleen, and tumor at Day 30 post injection. The mice carrying tumor received six types of materials including Au-half/all-OH in the pristine forms and their four derivatives coated with either ApoE or IgE. Data are presented as mean ± s.e.m. ($n = 3$). $p$ Values were calculated using multiple $t$ tests with no inner-group difference of statistical significance can be found. **b** Pathological analysis of major organs after a 7 and 30 days of exposure to ApoE-coated Au-all-OH. The tissue sections were stained with H&E. **c** Changes in the levels of biomarkers that would indicate a compromised tissue function. ALT alanine aminotransferase, TP total proteins, TBIL total bilirubin, PLT platelet, CRE creatinine, BUN blood urea nitrogen. Data are presented as mean ± s.e.m. ($n = 3$)

The all-atom MD simulations were performed by using the Gromacs 5.0.4 package[63] with OPLS-AA force field[64]. The TIP3P water model was chosen. The force-field parameters for the graphene were obtained from work by Tang et al.[65], and the force-field parameters for the Au surface and alkanethiol chains were obtained from previous work[66]. The particle mesh Ewald method was used to calculate the long-range electrostatic interactions, whereas the LJ interactions were treated with a cutoff distance of 1.2 nm. LINCS was applied to constrain the bond lengths within the solute. Periodic boundary conditions were applied in all three directions. The energy minimization was firstly performed for about 50,000 steps. After energy minimization, the system with the graphene sheet/Au surface and the protein positionally restrained was equilibrated for 1 ns at a constant pressure of 1 bar and a temperature of 298 K with Berendsen coupling methods[67]. Then, the restriction of the graphene sheet and protein was turned off, and the system was performed in the NVT ensemble at 298 K for 200 ns. The integration time step was 2 fs. Each system was simulated six times by changing the speed seed. To quantitatively depict the adsorption behavior of protein onto graphene, the CSA

between protein and graphene was introduced, which was defined as follows

$$CSA = \left( SASA_{pro} + SASA_{gra} - SASA_{pro+gra} \right)/2, \qquad (1)$$

where $SASA_{pro}$, $SASA_{gra}$, and $SASA_{pro+gra}$ represented the solvent accessible surface area of protein, graphene, and protein–graphene complex, respectively. The VMD software was used for trajectory visualization and analysis.

**In vivo distribution profiles.** We established three types of subcutaneous tumor models. For those grown from cancer cells, either H22 cells (mouse hepatocarcinoma cell line) or EMT-6 cells (mouse breast cancer cell line) were injected into the left flank of mice ($5 \times 10^6$ cells). The tumor was allowed to grow for about 5–8 days to reach a rough size of 100 mm³. Next, the tumor-bearing mice were randomly divided into six groups that received a systematical administration of ApoE/IgE-decorated G-all/half-OH and their pristine counterparts. Of note, the graphene used here was pre-labeled with gadolinium ions as mentioned previously[68] to make

the quantification of graphene feasible using ICP-MS (Agilent 7500ce, Agilent, United States). Next, they were functionalized with proteins. The dose regimen was fixed at 10 mg (in terms of graphene) for each mouse. The mice were sacrificed at 1, 4, 12, 24, and 48 h post injection, with their blood samples as well as liver, spleen, lung, and tumor tissues collected. These tissues were subjected to repeated frozen-thaw treatments, followed by digestion in boiling aqua regia ($HNO_3$: HCl, 1:3, vol/vol).The content of Gd was used to reflect the amount of graphene enriched in corresponding tissue using ICP-MS. For the assessment of Au nanosphere distribution, in vivo tests were conducted under identical condition except that the graphene was replaced by different types of 50-nm Au nanospheres ($50 \, mg \, kg^{-1}$ weight).

For the dual-labeling, Alexa Fluor-633-labeled protein was used for the passivation of Gd(III)-labeled graphene. The original fluorescence intensity and the content of Gd of the sample were measured with fluorescence spectra and ICP-MS, respectively, with their ratio calculated and leveraged to estimate the loss of original protein corona. In this case, the injection of graphene was conducted as did for the analysis of biodistribution. The mice were next sacrificed at 24 h post injection, with the collected blood and tissues divided into two parts. One-half of the sample was analyzed with ICP-MS to quantify the amount of graphene itself. Other half was minced via the 400-mesh screen, with the harvested single cell suspension stored at $-80 \, °C$ overnight in order to disrupt the cell integrity. Afterward, the cell suspension was centrifuged ($3000g \times 10 \, min$) to get rid of cell debris. We analyzed the fluorescence intensity of the supernatant and recalculated the ratio of [dye]/[graphene]. Thereafter, the loss of original protein corona can be reflected by the changes in such a ratio after the injection. To understand the preservation of coated proteins after 30 days, Au nanospheres carry either ApoE or IgE were injected into the tumor bearing mice (in this case, to keep the mice alive during the observation period, the original volume of tumor at the start point was set to $50 \, mm^3$, half of the value of previous biodistribution analysis). The tumor bearing mice were sacrificed 30 days later, and the content of gold remained in tumor tissue was analyzed by using ICP-MS, with the value of mice received pristine Au-all/half-OH served as control for comparison.

**Magnetic imaging**. Iron oxide was prepared using a co-precipitation-based method[69]. They were prepared freshly, and dispersed in phosphate buffer saline (pH ~7.2). Into the suspension we next introduced the proteins (final concentration: $10 \, mg \, mL^{-1}$) to fully passivate the surface of nanomaterials. Next, healthy mice received different types of $Fe_3O_4$ nanoparticles through tail vein injection, followed by MR imaging at 8 h later. In this case, the $T_2$-weighted MR images were collected on 7T ClinScan modality (Bruker).

**Statistics**. Triplicate data were analyzed with multiple $t$ tests using GraphPad Prism software (version 7.0); the significance level was set at $p < 0.05$. Significant statistical differences are indicated by asterisks in corresponding figures.

**Reporting summary**. Further information on research design is available in the Nature Research Reporting Summary linked to this article.

## Data availability
The authors declare that the data supporting the findings of this study are available in the article and Supplementary Information. Raw data or more information supporting the conclusion are available from the corresponding authors upon reasonable request.

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

## Acknowledgements

This work is supported by the National Natural Science Foundation of China (Nos. 11874045, 21604060, 11474155, and 11774147), Start-up funds of Nanjing Medical University to D.H., and Fundamental Research Funds for the Central Universities (0214-14380452). We are grateful to the High Performance Computing Center (HPCC) of Nanjing University for doing the numerical calculations in this paper on its blade cluster system.

## Author contributions

H.-M.D., D.H. and Y.-Q.M. conceived the idea and co-designed the simulation/experiment. X.L. performed the simulation, Y.-S.Y participated in performing the simulation. D.H. performed the in vitro and in vivo assessments with the assistance of P.X. H.-M.D., D.H., X.L. and Y.-Q.M. analyzed the data and co-wrote the paper. All authors have proof-read the paper and approved its publication in the present form.

## Competing interests

The authors declare no competing interests.
