## [Peer Review File · Nature Communications]

Reviewers' comments:

Reviewer #1 (Remarks to the Author):

In the manuscript by Lu et al. entitled "Tailoring the component of protein corona via simple chemistry: A computational and experimental investigation" the authors use graphene for correlating computational and experimental data.

First - although I would like to note that I am not a native speaker – I would feel that the language is seems to be awkward. Sentences like "...their hydrophobic residues are embedded inner the structure." or "...the protein should grow flattered" need to be corrected IMHO.

The citation of literature is adequate.

A major concern is that although graphene is a great material I do not see why it should be relevant to be used in vivo, especially intravenously. The aspect ratio of this 2D material prevents it use and may only be relevant for surface coatings of e.g. implants.

Only selected proteins were absorbed: are these relevant for graphene? Are they the most abundant proteins found on graphene? The authors should cite relevant literature to make their choice more understandable to the reader.

For PG, G-half-OH, G-all-OH: it is not surprising that there are differences. At least it would be needed to correlate these to findings for adsorption on silica, polystyrene or even better starch or protein nanocarriers.

The authors used fluorescently labelled proteins. I would caution or at least want them to discuss that labelling may change the adsorption behavior of proteins.

There is no added value by the in vivo studies IMHO. Graphene sheets are due to their aspect ratio are not to be used in vivo.

It is unclear how stable the Gd-labelling of graphene is. The author may have just quantified the free Gd. The authors need to present data that demonstrates that Gd is still bound to the graphene in vivo or at least in physiological, protein containing conditions.

In Fig. 7: after 1 h the start at 60% is much lower for IgG than for ApoE or pristine. Also ApoE is not overwhelmingly better than pristine graphene. The authors should report this more distinctly. Als this again demonstrates that a detailed protein corona analysis would be helpful.

In essence I would recommend this paper for major revision or even publishing it in another journal.

Reviewer #2 (Remarks to the Author):

In the manuscript "Tailoring the component of protein corona via simple chemistry: A computational and experimental investigation" Lu et al. reports a promising and facial strategy: ApoE-pre-coated graphene to prolong its blood circulation and improve the bio-distribution of the graphene materials in mouse tumor model as well. The paper computationally and experimentally investigated the impact of surface hydrophilicity (hydroxyl groups) on the interaction between HSA, IgE and ApoE respectively.

The protein corona formation on polymeric nanoparticles has shown its impact on prolonged blood circulation in *Nat Commun.* 2017;8(1):777. As for graphene, the protein corona formation has been demonstrated in some researches. (for instance: *ACS Nano.* 2015;9(6):5713-24. *Nat Commun.* 2018; 9: 1577. *Environ Sci Technol.* 2018. doi: 10.1021/acs.est.8b03451.) Though readers who are focusing on the graphene-based nanomedicine will feel interested in this strategy, following issues have to be properly addressed before being considered for publication:

1. As the authors claimed 'tailoring the component of protein corona via simple chemistry to extend the nanomaterials' blood circulation', is this strategy also applicable on other nanomaterials other than graphene?

2. The authors modulated surface hydrophilicity via controlling the content and distribution of hydroxyl groups, however, did the hydroxyl groups influence the surface charge of the graphene? Will the surface charge have impact on the nanomaterial-protein corona interaction? The surface charge of the materials needs to be confirmed. The analysis of their functional groups should also be included.

3. As demonstrated by many literatures, the biodistribution of nanomaterials is also influenced by their size. How about the size distribution of PG, G-ALL-OH and G-half-OH in this research? Did the protein corona change the physicochemical properties of materials, such as size and hydrophilicity, which in turn affect the clearance rate accordingly?

4. In the manuscript, the secondary structure contents of HSA, IgE and ApoE were calculated under MD simulation, were there any statistically significant differences among these different substrates? In addition, more experimental results need to be present, eg. circular dichroism spectroscopic measurement and AFM characterization.

5. More data is needed to confirm that ApoE-pre-coated-nanomaterial can lower the adsorption of the opsonins and recognition by reticuloendothelial system cells. Fig.8 analyzed the loss of original protecting protein layers but this is not enough. For example, since the proteins' morphological structure was influenced by the absorption of graphene materials, were the proteins still able to keep their function completely? Did ApoE/IgE -pre-coated-nanomaterial further interact with other proteins in the blood circulation? Was there any activation of complement system in vivo?

Other remarks,

6.Results are lack of statistical analysis or description in detail, such as Fig 7.

7.When representing the biodistribution profiles of the graphene in Fig 7, further comparisons are needed to explicitly support their enhanced accumulation in tumor and the decrement in liver, spleen and lungs. Moreover, is the passivation of ApoE for the graphene more beneficial in short-term or long-term clearance?

8.There are quite a few sentences not easy to understand. For example, line 35: 'Though potent in vitro, for a particular class of nanomedicine, the observation of its in vivo failure is not rare and surprising.'

Line 51: 'Meanwhile, recent evidence revealed that the elimination of protein adsorption is not necessarily the best approach if the therapeutic potential of nanomedicine is to be fully explored.'

Below we would like to point-by-point address the concerns from the reviewers.

Reviewer #1

Recommendation: Major revision as noted.

1) “First - although I would like to note that I am not a native speaker – I would feel that the language is seems to be awkward. Sentences like “...their hydrophobic residues are embedded inner the structure.” or “..the protein should grow flattered” need to be corrected”

Response: We thank the reviewer for his/her careful examination of our work, we have extensively revised the main text to ensure that it delivers a clear picture of our study. To further improve the language use, the revised manuscript has been double-checked by a professional science editor from Springer Nature before resubmission (Key: B396-D5A9-C4AC-39EC-D9AB). We hope the revised manuscript, in terms of flow and language use, can now meet the criteria of publication in *Nature Communications*.

2) “A major concern is that although graphene is a great material I do not see why it should be relevant to be used in vivo, especially intravenously. The aspect ratio of this 2D material prevents it use and may only be relevant for surface coatings of e.g. implants.”

Response: We appreciate the reviewer for raising an important point. We rewrote the introduction part by adding more details about how the choice of nanomaterials were made here. Overall, graphene is best known for the capacity as a vehicle for delivering hydrophobic therapeutics. Some recent reviewer articles have nicely summarized the progress in this field, and a latest, comprehensive one was cited here as **ref [42]** to strength our argument.

3) “Only selected proteins were absorbed: are these relevant for graphene? Are they the most abundant proteins found on graphene? The authors should cite relevant literature to make their choice more understandable to the reader.”

Response: We thank the reviewer for his/her valuable suggestions. Firstly, as indicated by a recent study (*Nat. Commun.* **2018**, *9*, 1577), the lipoproteins, immunoglobulins, and albumins are the most abundant types of proteins adsorbed on the graphene nanoflakes. Moreover, we also carried out a proteomic analysis to assess the protein corona of graphene as well as its derivatives. In brief, we let those materials differ in surface functionalization exposed to fresh harvested mouse serum wherein a library of proteins co-existed. Next, the made up of the immobilized corona, namely the hard corona, was assessed by LC-MS analysis. As shown below (**Supplementary Fig. 1**), enrichment of complementary system-related proteins, lipoproteins, and immunoglobulins was observed. As several nice publications have already reported ways to mitigate the passivation by complementary proteins like C3 (*Nat. Nanotech.* **2018**, *13*, 862-869; *Nat. Nanotech.* **2016**, *11*, 372-377), our attention was moved to the other two types of proteins, among which ApoE (Apolipoprotein E) and IgE

(Immunoglobulin E) were selected given their critical functions in fat metabolism and immunity, respectively. Apart from these two types, considering the great abundance in serum, HSA (human serum albumin) was also chosen. Altogether, proteins were selected on the basis of a concurrent consideration of abundance and biological implications. We understand that our previous interpretation might be too rough to be understandable. We thus extensively revised this part, with a highlight of rationales behind the selection of these proteins.

Supplementary Fig. 1 Proteomic analysis showing the composition of protein corona encasing G-half/all-OH and PG incubated with serum. Of note, the pie-chart was generated based on the number of recognized proteins belonging to each of the seven categories including lipoproteins, coagulation, tissue leakage, immunoglobulins, acute phase, complementary system, and other types.

4) “For PG, G-half-OH, G-all-OH: it is not surprising that there are differences. At least it would be needed to correlate these to findings for adsorption on silica, polystyrene or even better starch or protein nanocarriers.”

Response: We thank the reviewer for the constructive suggestions. We have thoroughly checked relevant publications about those materials recommended by the reviewer. Unfortunately, none of these ones have well established force field that allows us to perform the computer simulation. Without modelling, we cannot compare the binding of proteins with different substrates at the molecular level. To this end, after a careful examination of materials favorably exploited for biological applications, we selected gold (Au) as the choice. Gold nanomaterial is one of the most popularly vehicles being employed for diagnosis or therapy owing to its superior biocompatibility and ease of surface functionalization by harnessing the Au-S chemistry. We tailored its availability of hydroxyl groups by employing a silver-assisted, ligand exchange method we developed recently (*J. Am. Chem. Soc.* **2018**, *140*, 11898-11901). Briefly, aqueous soluble, 50-nm Au nanospheres were synthesized followed by a deposition of a thin layer of silver to exclude the previous capping agents. During the removal of silver shell, pristine Au surface was exposed, which allowed foreign ligands to passivate almost

stoichiometrically. As such, we synthesized the Au nanoparticles carrying full or halved hydroxyl groups. To help appreciate this process, a schematic illustration was added (Fig. 6a). Corresponding *in vitro* findings (Fig. 6b) correlates well with that of graphene and was further explained by the computation at molecular level (Fig. 6 c,d and Supplementary Fig. 3-5). This strongly suggest that it is the surface hydroxyl groups, rather the material nature, modulates the manner of interaction with serum proteins. Besides, *in vivo* biodistribution profiles (Fig. 7b) and long-term toxicity (Fig. 10) of Au were also evaluated.

Fig. 6 Profiles of protein adsorbed on Au nanoparticles. **a** Schematic illustrating the step-wise introduction of foreign ligands with the assistance of Ag. I: Deposition of a thin layer of Ag; II: Removal of Ag alongside with the passivation with ligands. **b** *In vitro* analyses showing the number of proteins adsorbed on the graphene. Data are presented as mean \pm s.e.m. ($n=3$). P values were calculated using multiple t tests ($***p<0.001$). **c** Typical final snapshots of the adsorption of three types of proteins onto the Au nanoparticle (yellow) with different surface modifications. The hydrophobic residues within 0.5 nm distance from the surface are represented as CPK spheres. Water molecules and ions are omitted for clarity. **d** The CSA profiles of proteins adsorbed on different Au nanospheres. Data are presented as mean \pm s.e.m. ($n=5$). P values were calculated using multiple t tests ($*p<0.05$). For a full spectrum of snapshots and interaction energy profiles, please refer to Supplementary Fig. 3-5.

Supplementary Fig. 3 The simulation results of the interaction of HSA with the Au nanoparticle. **a** Typical snapshots of the initial and final structures of the adsorption of HSA onto the surface (yellow) with different surface modifications. From the side and top views, hydrophobic residues within 0.5 nm distance from the surface are represented as CPK spheres. Water molecules and ions are omitted for clarity. **b, c** The contact surface area (CSA) and the Lennard-Jones (LJ) and Coulomb (Coul) interaction energy between HSA and the Au nanoparticle with different surface modifications, respectively. Data are presented as mean \pm s.e.m. (n=5). *P* values were calculated using multiple t tests (**p*<0.05).

Supplementary Fig. 4 The simulation results of the interaction of IgE with the Au nanoparticle. **a** Typical snapshots of the initial and final structures of the adsorption of IgE onto the surface (yellow) with different surface modifications. From the side and top views, hydrophobic residues within 0.5 nm distance from the surface are represented as CPK spheres. Water molecules and ions are omitted for clarity. **b, c** The contact surface area (CSA) and the Lennard-Jones (LJ) and Coulomb (Coul) interaction energy between IgE and the Au nanoparticle with different surface modifications, respectively. Data are presented as mean \pm s.e.m. (n=5). *P* values were calculated using multiple t tests (**p*<0.05).

Supplementary Fig. 5 The simulation results of the interaction of ApoE with the Au nanoparticle. **a** Typical snapshots of the initial and final structures of the adsorption of ApoE onto the surface (yellow) with different surface modifications. From the side and top views, hydrophobic residues within 0.5 nm distance from the surface are represented as CPK spheres. Water molecules and ions are omitted for clarity. **b, c** The contact surface area (CSA) and the Lennard-Jones (LJ) and Coulomb (Coul) interaction energy between ApoE and the Au nanoparticle with different surface modifications, respectively. Data are presented as mean \pm s.e.m. ($n=5$). P values were calculated using multiple t tests ($*p<0.05$).

Fig. 7b Biodistribution of Au-half/all-OH in four tissues including liver, spleen, lungs, and kidneys in addition to blood, respectively. Data are presented as mean \pm s.e.m. ($n=3$ at each time point) as a function of time. P values were calculated using multiple t tests ($***p<0.001$, $**p<0.01$, $*p<0.05$).

Fig. 10 Long-term toxicity. **a** Content of Au element remained in blood and three tissues including liver, spleen, and tumor at Day 30 post injection. The mice carrying tumor received six types of materials including Au-half/all-OH in the pristine forms and their four derivatives coated with either ApoE or IgE. Data are presented as mean \pm s.e.m. ($n=3$). *P* values were calculated using multiple t tests with no inner-group difference of statistical significance can be found. **b** Pathological analysis of major organs after a 7 and 30 days of exposure to ApoE-coated Au-all-OH. The tissue sections were stained with H&E. **c** Changes in the levels of biomarkers that would indicate a compromised tissue function. ALT: alanine aminotransferase; TP: total proteins; TBIL: total bilirubin; PLT: platelet; CRE: creatinine; BUN: blood urea nitrogen. Data are presented as mean \pm s.e.m. ($n=3$).

Moreover, to test the idea of using pre-ApoE coating to extend nanomaterials' blood circulation, (apart from the Au nanomaterial) we also choose the Fe_3O_4 nanomaterial, a commercialized product for contrast-enhanced magnetic resonance imaging (**Supplementary Fig. 7**). Like that for graphene, a pre-coating here rewarded Fe_3O_4 reduced liver uptake as reflected by the weaker contrast in tissue relative to counterparts in pristine form or carrying IgE before blood entry.

Supplementary Fig. 7 **a** T_2 -weighted magnetic resonance imaging results. Images of liver tissue of mice received different nanomaterials were collected at 8 h post the injection. **b** Quantification of imaging results by using the signal-to-noise ratio (SNR) to reflect the content of enhancement in each case. Data are presented as mean \pm s.e.m. (n=5). P values were calculated using multiple t tests (* p <0.05).

5) “The authors used fluorescently labelled proteins. I would caution or at least want them to discuss that labelling may change the adsorption behavior of proteins.”

Response: We appreciate the reviewer for raising an important point. As it is hard to quantify tiny amount of proteins, we replaced the pristine proteins with fluorescent counterparts in some cases. The reviewer would be glad to find that other groups are doing things in the same way. (please refer to *ACS Nano* **2016**, *10*, 10471-10479). However, we totally agree that the introduction of dye into the protein remodel its interaction with cells. We have highlighted this likelihood in our revised manuscript (Page 5) which reads as:

“Of note, the introduction of a fluorescent dye into the protein might slightly change the manner of its interaction with substrates but not to a great extent⁴⁷. To avoid misleading results, the comparisons here and thereafter were all made between groups using the same type of proteins, that is, fluorescent or non-fluorescent.”

6) “There is no added value by the *in vivo* studies IMHO. Graphene sheets are due to their aspect ratio are not to be used *in vivo*.”

Response: The propose of conducting *in vivo* analyses was to verify if the pre-adsorption with proteins can help tune their pharmacokinetics. In this case, we raised two important findings that has rarely been reported before. First, it was proved that decoration with ApoE proteins can effectively extend its blood circulation, and the stability of such pre-formed corona can be adjusted by tuning the material surface property. A replacement of ApoE with IgE saw an opposite effect. Underlying their action, we notice that ApoE behaves like dysopsonins that help resist the coating with complementary proteins any further, whereas IgE functions in a way similar to typical opsonins.

Another intriguing result is the tissue-relevant retention of previously coated proteins. Almost all nanomaterial carrying proteins would experience loss of decoration during blood circulation, which can be taken as a common sense. However, it has never expected that to which extent these

nanomaterials preserve the coating can be associated with their tissue preference. In brief, we found that either a greater loss of ApoE or a better preservation of IgE can result in increased uptake by mononuclear phagocyte system-related tissues including liver and spleen. We further proved that such an observation was not occasional that linked only to certain types of tumors. Three types of tumors including one derived from human patient were studied herein. All findings indicate that such a phenomenon is universal, and at least partially explain why roughly the same nanomaterials differ in fate *in vivo*.

Recent evidences suggest that the replacement of 3-D materials with 2-D counterparts (the family graphene belongs to) can bring additional merits. The most significant one of them is the ultra-high drug loading capacity as 2-D materials tend to exhibit high surface area. Additional benefits including the novel optical property of capability to sensitize ionizing radiation have also been reported as summarized in an up-to-date review article (*Adv. Mater.* **2019**, 1902333). Meanwhile, we totally understand the reviewer's concern about the feasibility of using 2-D nanomaterials as nanomedicine. As such, we did add another two types of nanomaterials (Au nanospheres and Fe₃O₄ nanoparticles) that have been favorably employed for many purposes with superior biocompatibility. Based on the preliminary findings, we conclude that a tuned protein corona as well as the proposed strategy to extend blood circulation is tightly linked to hydrophilicity, and was almost independent on the material itself. We have emphasized this point in our revised paper.

7) *“It is unclear how stable the Gd-labelling of graphene is. The author may have just quantified the free Gd. The authors need to present data that demonstrates that Gd is still bound to the graphene in vivo or at least in physiological, protein containing conditions.”*

Response: We thank the reviewer for the valuable suggestion. We have assessed the stability of Gd tag over time. To better mimic the condition graphene faced *in vivo*, we selected fresh mouse serum. Briefly, the graphene conjugated with Gd complex was suspended in fresh mouse serum. In a course of 7 days, we collected part of the medium each day and get the graphene harvested through centrifugation, with the content of free Gd analyzed by using inductively coupled plasma-mass spectrometer (ICP-MS). As shown below (also as **Supplementary Fig. 6** in our revised paper). We found that the labeling of graphene with Gd here was stable for at least 3 days, with the loss of Gd accounting to less than 2% of the total conjugation. Meanwhile, as our biodistribution analyses were finished within 72 h after the administration of nanomaterials, it is reasonable to see the measured Gd contents faithfully reflecting the content of graphene. Similar labeling strategy has also been used in previous study (please refer to *Nat. Nanotechnol.* **2015**, *10*, 619-623), which has been cited here as **ref [56]** alongside a paragraph of discussion on the stability of labeling.

Supplementary Fig. 6 Integrity of Gd-tag. The Gd-labeled graphene was incubated with fresh mouse serum, and the content of free Gd formed as a consequence of premature detachment was analyzed by inductively coupled plasma-mass spectrometer (ICP-MS). Three independent tests were conducted during a period of 7 days, with each result shown individually.

8) *“In Fig. 7: after 1 h the start at 60% is much lower for IgG than for ApoE or pristine. Also ApoE is not overwhelmingly better than pristine graphene. The authors should report this more distinctly. Als this again demonstrates that a detailed protein corona analysis would be helpful.”*

Response: For the low blood availability of IgE-functionalized nanomaterials, we speculated that the IgE accelerated the blood clearance of graphene. As the reviewer recommended, we were glad to see deeper into the protein corona formed on graphene by using proteomics. Of note, the composition of protein corona formed *in vivo* is beyond our capacity to characterize since there exists no suitable way to enrich nanomaterials only without introducing impurities. Through centrifugation, it is hard to tell apart the proteins adsorbed on graphene or originated from tissue fragments. As such, what to our best can be done so far was a comprehensive analysis of protein corona formed on graphene that received freshly prepared mouse serum. Proteomic results (**Fig. 8a**) reveal that IgE provokes the complementary system that later tags the graphene as invading species. Considering that professional phagocytes can act soon after the activation of complementary system, it is not surprising to see the IgE-coated materials experience a rapid clearance from blood.

We also notice that the protein corona encasing IgE-functionalized graphene is rich in immunoglobulins (Ig). Given their cascade-like working fashion, there remains another possibility that those Ig proteins drives the stacking of 2-D graphene sheets into aggregates featuring reduced half-life. This is partly supported by the observation that replacement of graphene with Au nanospheres (**Fig. 8c**), while leaving other parameters unchanged, led to slightly extended blood circulation.

For pre-coating with ApoE, it benefits graphene and Au nanomaterials to different extents. In graphene case (taking G-half-OH as an example), we have observed an extension of half-life from 1.7 h (pristine form) or 2.3 h (IgE-coated) to 7.6 h. For Au nanospheres, an ever-greater extension in circulation time was observed. Taking the Au-half-OH as another example, the coating with ApoE added a 3.5 h to the

half-life of materials. These findings together serve as evidence supporting the enormous practical value of using ApoE for pre-coating.

Fig. 8 Biological implications of pre-adsorbed proteins. **a** Proteomic analysis showing the composition of protein corona encasing G-half/all-OH and PG pre-coated with ApoE/IgE. Of note, the pie-chart was generated based on the number of recognized proteins belonging to each of the seven categories including lipoproteins, coagulation, tissue leakage, immunoglobulins, acute phase, complementary system, and other types. **b** Macrophage uptake of dye-labeled G-all-OH and counterparts coated with ApoE or IgE. Lysosome compartment and nucleus were stained to guide the eyes. **c** Quantified uptake by macrophages of Au-all-OH and protein-coated counterparts. Data are presented as mean ± s.e.m. (n=3). P values were calculated using multiple t tests (***) $p < 0.001$.

Reviewer #2

Recommendation: Publish after Major revisions.

“In the manuscript “Tailoring the component of protein corona via simple chemistry: A computational and experimental investigation” Lu et al. reports a promising and facial strategy: ApoE-pre-coated graphene to prolong its blood circulation and improve the bio-distribution of the graphene materials in mouse tumor model as well. The paper computationally and experimentally investigated the impact of surface hydrophilicity (hydroxyl groups) on the interaction between HSA, IgE and ApoE respectively.

The protein corona formation on polymeric nanoparticles has shown its impact on prolonged blood circulation in Nat Commun. 2017;8(1):777. As for graphene, the protein corona formation has been demonstrated in some researches. (for instance: ACS Nano. 2015;9(6):5713-24. Nat Commun. 2018; 9: 1577. Environ Sci Technol. 2018. doi: 10.1021/acs.est.8b03451.) Though readers who are focusing on the graphene-based nanomedicine will feel interested in this strategy, following issues have to be properly addressed before being considered for publication:”

We are grateful for the reviewer’s overall appreciation of our work. We are aware of these nice

publications and cited them in our revised manuscript. Below we would like to address your concerns point-by-point.

1) “As the authors claimed ‘tailoring the component of protein corona via simple chemistry to extend the nanomaterials’ blood circulation’, is this strategy also applicable on other nanomaterials other than graphene?”

Response: We thank the reviewer for raising an important point. We managed to extend our reported strategy to another two favorably employed materials, Au and Fe₃O₄ nanoparticles (**Fig. 6, Fig. 7b, Fig. 8, Fig. 10, and supplementary Fig. 7**, respectively). They were selected when considering their superior compatibility and enormous practical value for diagnosis or disease therapy purposes. Furthermore, as the force field of gold element is well established, the manner of Au nanomaterials-protein interaction was also assessed by using computer simulation in a way similar to what we have done for graphene (**Fig. 6c, d**). The simulation results coincided with that of experimental observations (**Fig. 6b**). Likewise, encasement with ApoE also effectively helps Fe₃O₄ escape from the being captured by Kupffer cells that are abundant in liver (**supplementary Fig. 7**). Combining with the proteomic results, our proposed strategy is confirmed to be feasible in shaping a protein corona rich in dysopsonin that benefits extended blood circulation. Such an action exhibits little to no dependence on the material itself. All these findings were discussed in details in our revised manuscript, and we hope the reviewer would be satisfied with the changes we made.

Fig. 6 Profiles of protein adsorbed on Au nanoparticles. **a** Schematic illustrating the step-wise introduction of foreign ligands with the assistance of Ag. I: Deposition of a thin layer of Ag; II: Removal of Ag alongside with the passivation with ligands. **b** In vitro analyses showing the number of proteins adsorbed on the graphene. Data are presented as mean \pm s.e.m. ($n=3$). P values were calculated using multiple t tests ($***p<0.001$). **c** Typical final snapshots of the adsorption of three types of proteins onto the Au nanoparticle (yellow) with different surface modifications. The hydrophobic residues within 0.5 nm distance from the surface are represented as CPK spheres. Water molecules and ions are omitted for clarity. **d** The CSA profiles of proteins adsorbed on different Au nanospheres. Data are presented as mean \pm s.e.m. ($n=5$). P values were calculated using multiple t tests ($*p<0.05$). For a full spectrum of snapshots and interaction energy profiles, please refer to **Supplementary Fig. 3-5**.

Fig. 7b Biodistribution of Au-half/all-OH in four tissues including liver, spleen, lungs, and kidneys in addition to blood, respectively. Data are presented as mean \pm s.e.m. ($n=3$ at each time point) as a function of time. P values were calculated using multiple t tests (** $p < 0.001$, * $p < 0.01$, $p < 0.05$).

Fig. 8 Biological implications of pre-adsorbed proteins. **a** Proteomic analysis showing the composition of protein corona encasing G-half/all-OH and PG pre-coated with ApoE/IgE. Of note, the pie-chart was generated based on the number of recognized proteins belonging to each of the seven categories including lipoproteins, coagulation, tissue leakage, immunoglobulins, acute phase, complementary system, and other types. **b** Macrophage uptake of dye-labeled G-all-OH and counterparts coated with ApoE or IgE. Lysosome compartment and nucleus were stained to guide the eyes. **c** Quantified uptake by macrophages of Au-all-OH and protein-coated counterparts. Data are presented as mean \pm s.e.m. ($n=3$). P values were calculated using multiple t tests (** $p < 0.001$).

Fig. 10 Long-term toxicity. **a** Content of Au element remained in blood and three tissues including liver, spleen, and tumor at Day 30 post injection. The mice carrying tumor received six types of materials including Au-half/all-OH in the pristine forms and their four derivatives coated with either ApoE or IgE. Data are presented as mean \pm s.e.m. ($n=3$). *P* values were calculated using multiple t tests with no inner-group difference of statistical significance can be found. **b** Pathological analysis of major organs after a 7 and 30 days of exposure to ApoE-coated Au-all-OH. The tissue sections were stained with H&E. **c** Changes in the levels of biomarkers that would indicate a compromised tissue function. ALT: alanine aminotransferase; TP: total proteins; TBIL: total bilirubin; PLT: platelet; CRE: creatinine; BUN: blood urea nitrogen. Data are presented as mean \pm s.e.m. ($n=3$).

Supplementary Fig. 7 T_2 -weighted magnetic resonance imaging results. Images of liver tissue of mice received different nanomaterials were collected at 8 h post the injection. **b** Quantification of imaging results by using the signal-to-noise ratio (SNR) to reflect the content of enhancement in each case. Data are presented as mean \pm s.e.m. ($n=5$). *P* values were calculated using multiple t tests ($*p<0.05$).

2) *“The authors modulated surface hydrophilicity via controlling the content and distribution of hydroxyl groups, however, did the hydroxyl groups influence the surface charge of the graphene? Will the surface charge have impact on the nanomaterial-protein corona interaction? The surface charge of the materials needs to be confirmed. The analysis of their functional groups should also be included.”*

Response: We have analyzed the zeta-potential of graphene and its derivatives. It was found that the introduction of hydroxyl groups onto graphene only minorly affect the net charge (-0.24 ± 0.1 mV, -0.67 ± 0.2 mV, and -0.33 ± 0.1 mV for PG, G-all-OH, and G-half-OH). Of note, despite the variation in surface functionalization, all materials are bearing weakly negative charges, promoting us to believe that their interaction with proteins as driven by electrostatic force is not expected to vary markedly. According to your suggestion, we have also added a brief discussion on this point, which can be found on Page 6, which reads:

“Changes in both the net charge and surface chemical properties can be behind differences in attraction to proteins. After analysing the zeta-potential of G-all/half-OH and their pristine counterpart, we noticed that they all carried slightly negative net charges (-0.24 ± 0.1 mV, -0.67 ± 0.2 mV, and -0.33 ± 0.1 mV for PG, G-all-OH, and G-half-OH, respectively), revealing that a simple electrostatic force-driven interaction with proteins was not the case here.”

To prepare both graphene and Au nanomaterials differ in available hydroxyl groups, we followed published protocols, which included detailed characterization that promised the feasible in tailoring the surface functionalization. We have cited these papers respectively as **ref [44]** and **[55]** for the interested readers' reference.

3) *“As demonstrated by many literatures, the biodistribution of nanomaterials is also influenced by their size. How about the size distribution of PG, G-ALL-OH and G-half-OH in this research? Did the protein corona change the physicochemical properties of materials, such as size and hydrophilicity, which in turn affect the clearance rate accordingly?”*

Response: We thank the reviewer for raising an important yet missing point, we have analyzed the size distribution of graphene and its derivatives using dynamic light scattering (DLS). Despite extensive surface modification has been made to graphene, only a limited addition to mean hydrodynamic diameter (HD) was observed (from 97 nm for PG to 119 nm and 113 nm for G-all-OH and G-half-OH, respectively). We further characterized how the formation of protein corona made up with an individual type of proteins modulated the HD. As shown in **Fig. 1d**, taking G-all-OH as the model, we found that the passivation with IgE led to the least increasement of HD, but unexpectedly resulted in aggregation of G-all-OH to some extent. From the width of each curve, it is reasonable to say the coating with HSA and ApoE can still allow the nanomaterials to keep its dispersity. It is hard for us to estimate the change in hydrophilicity post the coating with proteins. Meanwhile, based on our later observation in proteomic analysis and *in vitro* uptake assay (for more details, please refer to our answer to your question #5), we believe the change in pharmacokinetics of coated nanomaterials (*i.e.*, rate of clearance) is more likely a consequence of different protein corona whose components are recruited by either ApoE or IgE.

Fig. 1 Structural information of Graphene and derivatives and their *in vitro* interaction with proteins. **a, b** Graphene sheet (cyan) with different surface modifications in the experiment and simulation. PG represents the pristine graphene, while G-all-OH stands for the graphene with the hydroxyl groups decorated onto both sides of the whole basal plane, and the hydroxyl groups on G-half-OH was roughly half of that exposed by G-all-OH, respectively. **c** AFM results of G-all-OH passivated with HSA, ApoE, or IgE, respectively. **d** Results of DLS showing the size changes post protein adsorption. **e** *In vitro* analyses showing the number of proteins adsorbed on the graphene. Data are presented as mean \pm s.e.m. (n=3). *P* values were calculated using multiple t tests (** $p < 0.001$, ** $p < 0.005$).

4) “In the manuscript, the secondary structure contents of HSA, IgE and ApoE were calculated under MD simulation, were there any statistically significant differences among these different substrates? In addition, more experimental results need to be present, eg. circular dichroism spectroscopic measurement and AFM characterization.”

Response: We have added the statistical details of our computational results. As shown in **Fig. 5a**, there indeed exist the statistically significant differences among these different substrates.

Taken the G-all-OH as an example, we have also analyzed the surface morphology post-adsorption with three types of proteins by using atomic force microscope as recommended by the reviewer. The result can be found in **Fig. 1c**. We notice that the HSA proteins adsorbed onto graphene tend to stay isolated, creating an island-like distribution, whereas the ApoE and IgE proteins are more likely to connect with adjacent counterparts to form an intact coating layer. We have briefly discussed the difference in surface morphology on Page 5, which reads:

“Taking G-all-OH as an example, the morphology of nanomaterials coated with different types of proteins was analysed by atomic force microscopy. In **Fig. 1c**, HSA molecules passivating the nanomaterials tended to stay isolated from each other, whereas adsorbed ApoE and IgE molecules appeared to be connected and created a thin layer-like corona.”

5) “More data is needed to confirm that ApoE-pre-coated-nanomaterial can lower the adsorption of the opsonins and recognition by reticuloendothelial system cells. Fig.8 analyzed the loss of original protecting protein layers but this is not enough. For example, since the proteins’ morphological structure was influenced by the absorption of graphene materials, were the proteins still able to keep their function completely? Did ApoE/IgE -pre-coated-nanomaterial further interact with other proteins in the blood circulation? Was there any activation of complement system *in vivo*?”

Response: We appreciate the reviewer for his/her constructive comments. We first conducted a proteomic analysis to better understand whether and more importantly how the pre-adsorbed proteins remodel the protein corona. In this case, the graphene derivatives differed in coated proteins were incubated with fresh mouse serum. After centrifugation, the composition of protein corona encasing the nanomaterials were assessed. As shown in **Fig. 8a** (Page 14 of the response letter), we notice that the ApoE help creates a protein corona rich in members from the lipoprotein family, leaving much less room for complementary protein to stay, especially in G-all/half-OH materials. By contrast, proteins belonging complementary system overwhelm the corona. As requested by the reviewer, we further assessed the recognition of nanomaterials by mononuclear phagocyte system. To better understand the role of protein corona, we let the pre-coated nanomaterials (both graphene and Au nanosphere) exposed to mouse serum in prior to their incubation with macrophages that functioned as professional phagocyte in RES. Fluorescence imaging (**Fig. 8b**) suggest that ApoE reduced the likelihood of recognition by macrophages for coated nanomaterials. Such an observation was also quantitatively supported by the ICP-MS results, in which uptake of Au nanospheres were analyzed (**Fig. 8c**). Taken together of the *in vitro* results and proteomic findings, we propose that the ApoE coating help extend the half-life of nanomaterials through remodeling the protein corona into one rich in components that are less recognizable by RES. Moreover, the preservation of original protein corona was analyzed in two additional tumor model including the patient-derived xenograft (PDX) model that resemble the pathological features of human cancer. We have added the results into **Fig. 9**, and it can be found that the findings in either PDX model or that featuring a high interstitial pressure all coincide with that of H22 tumor, revealing that the association between protein corona and the tendency of nanoparticles to enrich in certain tissue is independent of tumor type. We have briefly discussed this results in our revised manuscript.

We totally understand the concern raised by the reviewer that the adsorption of proteins onto the nanomaterials might risk compromising their functions to some extent. To verify this hypothesis, we took ApoE as an example, and performed additional simulations to compare the activity of the adsorbed proteins with that of the free protein. As shown in **Supplementary Fig. 2**, since the flexibility of the protein at the active sites decreased a lot (*i.e.*, the number of residues with negative ΔRMSF was great) post-adsorption in the case of PG, the biological function of ApoE in this case was compromised to some extent. Nevertheless, in the cases of G-all/half-OH, the number of residues with a positive ΔRMSF was so close to that of negative ΔRMSF , suggesting that the biological function of ApoE was well preserved. Thus, the presence of hydroxyl groups offers a promising way to minimize the denaturation effect exerted by graphene materials.

Fig. 9 Loss of original coating proteins. The amount of pre-adsorbed proteins on nanomaterials were quantified using ICP-MS at 24 h post the administration of G-all/half-OH, with the percentage of remainder capping layer calculated as described in the experimental section. Cell-dense tumor and high IFP stand for the xenograft grown from H22 cells and EMT-6 cells, respectively. The PDX model was established by transplanting the tissue sections harvested from breast cancer-suffering patients into nude mice. Data are presented as mean \pm s.e.m. (n=3). In the box plot, the upper and lower quartile as outlined by top and bottom boundary were divided by line showing the median value.

Supplementary Fig. 2 The RMSF (root mean square fluctuation) difference parameter Δ RMSF of the active/binding site (residues 136-150) of ApoE when adsorbed onto graphene sheet with different surface modifications (PG, G-half-OH, G-all-OH). Δ RMSF is used to characterize the change in flexibility of residues (compared to that in free state), i.e., Δ RMSF=(RMSF_{gra}-RMSF_{free})/RMSF_{free}, where RMSF_{free} is the RMSF of the C α atom of residues on the ApoE in free state (i.e., in solution), and RMSF_{gra} is the RMSF of the C α atom of residues on the ApoE adsorbed on PG/G-half-OH/G-all-OH. A positive value of Δ RMSF (i.e., Δ RMSF >0) indicates an increase in the flexibility of residues, while a negative value (i.e., Δ RMSF <0) indicates a decrease in the flexibility. The inserted table summarizes the increased or decreased numbers

of Δ RMSF in all cases. The more the increased number (of Δ RMSF), the more active the binding site is; the more the decreased number (of Δ RMSF), the less active the binding site is.

6) *“Results are lack of statistical analysis or description in detail, such as Fig 7.”*

Response: We have added corresponding statistical analysis and description into figures/figure captions according to your suggestion.

7) *“When representing the biodistribution profiles of the graphene in Fig 7, further comparisons are needed to explicitly support their enhanced accumulation in tumor and the decrement in liver, spleen and lungs. Moreover, is the passivation of ApoE for the graphene more beneficial in short-term or long-term clearance?”*

Response: We aware that the presentation of distribution profiles in tumor alongside that of liver and spleen can result in a seemingly less difference in tumor accumulation for various types of materials. As such, we remade the whole figures, and distribution profiles of relevant materials was plotted separately on the basis of organ. In **Fig. 7**, it can clearly be observed that the ApoE passivation facilitated the tumor entry of both graphene (**Fig. 7a**) and Au nanospheres (**Fig. 7b**, Page 14 of the response letter). Comparing to either pristine or IgE-coated nanomaterials, such an increasement is of statistical significance.

Besides, as the reviewer mentioned the potential long-term action of ApoE, we have also conducted a biodistribution analysis after 30 days post injection. In this case, as the Gd-complex we used to tag graphene become significantly vulnerable after 7 days *in vivo*, we thereby focus on Au nanospheres that require no foreign label as the material itself can be recognized readily by inductively coupled plasma-mass spectrometer (ICP-MS). Here, we noticed that there remained similar amount of Au nanospheres whose original coating were different (**Fig. 10a**, Page 15 of the response letter). As such, it is reasonable to speculate that the nanomaterials get rid of coated proteins over time. Given that the coated proteins could affect the fate of nanomaterials within 3 days post injection, it is more likely that their detachment takes place several days after the blood entry, which might take no longer than 4 weeks to accomplish. As in both Au nanospheres carrying a full extent or halved hydroxyl groups saw the same results, we envision that the gradual loss of original protein coating is promoted by the sophisticated niche, rather than difference in physiochemical property of nanomaterials. In addition, as reflected by the pathological and hematological results (**Fig. 10b, c**), no obvious change in tissue structure and absence of chronic inflammatory response reveal that the pre-coating with ApoE does not add to the cytotoxicity of nanomaterials.

8) *“There are quite a few sentences not easy to understand. For example, line 35: ‘Though potent in vitro, for a particular class of nanomedicine, the observation of its in vivo failure is not rare and surprising.’ Line 51: ‘Meanwhile, recent evidence revealed that the elimination of protein adsorption is not necessarily the best approach if the therapeutic potential of nanomedicine is to be fully explored.’”*

Response: We thank the reviewer for his/her careful examination of our manuscript. We have made extensive revision on the manuscript in terms of language use. In addition, the revised manuscript has been proof-read and corrected by professional editor from Spring Nature Group (Key: B396-D5A9-C4AC-39EC-D9AB). We believe the revised manuscript now gives a clear picture of findings with easy-to-understand interpretations.

We have also made some additional editorial changes to further enhance the quality of our paper. It is hoped the revised manuscript is now in the right format for publication in *Nature Communications*.

Sincerely yours,

Yu-qiang Ma

REVIEWERS' COMMENTS:

Reviewer #1 (Remarks to the Author):

The authors have done a great deal of work and have presented additional data as recommended as well as have been addressing concerns and added clarifications in the manuscript. Keeping also in mind that research is never giving all the answers to a set of questions I would now recommend that this paper is published in its present form. The paper is presenting interesting data in a concise form and adds to the discussions in the field.

Reviewer #2 (Remarks to the Author):

The authors have made great efforts on improving the manuscript. And my comments were addressed appropriately.